# Behavioral fingerprints predict insecticide and anthelmintic mode of action

Adam McDermott-Rouse[1,2,†], Eleni Minga[1,2,†], Ida Barlow[1,2] (iD), Luigi Feriani[1,2], Philippa H Harlow[3], Anthony J Flemming[3] & André E X Brown[1,2,*] (iD)

## Abstract

**Novel invertebrate-killing compounds are required in agriculture and medicine to overcome resistance to existing treatments. Because insecticides and anthelmintics are discovered in phenotypic screens, a crucial step in the discovery process is determining the mode of action of hits. Visible whole-organism symptoms are combined with molecular and physiological data to determine mode of action. However, manual symptomology is laborious and requires symptoms that are strong enough to see by eye. Here, we use high-throughput imaging and quantitative phenotyping to measure *Caenorhabditis elegans* behavioral responses to compounds and train a classifier that predicts mode of action with an accuracy of 88% for a set of ten common modes of action. We also classify compounds within each mode of action to discover substructure that is not captured in broad mode-of-action labels. High-throughput imaging and automated phenotyping could therefore accelerate mode-of-action discovery in invertebrate-targeting compound development and help to refine mode-of-action categories.**

**Keywords** anthelmintics; *C. elegans*; computational ethology; pesticide; phenotypic screen

**Subject Categories** Computational Biology; Methods & Resources; Pharmacology & Drug Discovery

**Mol Syst Biol. (2021) 17: e10267**

## Introduction

Invertebrate pests including insects, mites, and nematodes damage crops, decrease livestock productivity, and cause disease in humans. Nematodes alone infect over 1 billion people and lead to the loss of 5 million disability-adjusted life-years annually (Pullan *et al*, 2014). In livestock, they infect sheep, goats, cattle, and horses causing gastroenteritis that leads to diarrhea, reduced growth, and weight loss. Nematodes that parasitize crops have been estimated to cause

well over $100 billion in annual crop losses (Elling, 2013). Crop loss due to insects is measured in tens of metric megatons and is predicted to increase due to climate change (Deutsch *et al*, 2018). Compounds that kill or impair invertebrates are one of the primary means of defense in human and veterinary medicine and in crop protection. However, resistance is widespread in nematodes and insects and drives continuing efforts to discover new invertebrate-targeting compounds (Sparks & Nauen, 2015; Nixon *et al*, 2020).

To date, most currently approved treatments for infections in humans and livestock and for crop protection in the field have been discovered through phenotypic screens (Geary *et al*, 2015; Wing, 2020). That is, compounds are first screened for the ability to kill or impair a target species without any hypothesized molecular target. A critical problem is then determining hit compounds' mode of action, which is important for understanding resistance mechanisms, avoiding pathways where resistance is already common, and subsequent lead optimization. Despite advances in biochemical and genetic methods for determining mode of action, direct observation of the symptoms induced by compounds remains a key step in mode-of-action discovery (Wing, 2020). Because most insecticides and anthelmintics target the neuromuscular system, behavioral symptoms are a particularly important class of phenotypes to consider, but manual observation of behavior is time-consuming, insensitive to subtle phenotypes, and prone to inter-operator variability and bias (Garcia *et al*, 2010). We therefore sought to develop more automated and quantitative methods to do mode-of-action prediction from phenotypic screens of freely behaving invertebrates.

Pioneering work in zebrafish showed that behavioral fingerprints can be used to discover neuroactive compounds and that behavioral fingerprints correlate with compound mode of action (Kokel *et al*, 2010; Rihel *et al*, 2010; Laggner *et al*, 2011). However, this approach has not yet been applied to invertebrate animals—the targets of insecticides and anthelmintics—at a large scale. Furthermore, although previous zebrafish screens were high throughput, their spatial resolution was low and phenotypes were limited to activity levels in response to stimuli. Recent work in computational ethology has shown the power of moving beyond point representations of animal behavior to include information on posture (Anderson & Perona, 2014; Egnor & Branson, 2016; Berman, 2018; Brown & de

1   MRC London Institute of Medical Sciences, London, UK
2   Faculty of Medicine, Institute of Clinical Sciences, Imperial College London, London, UK
3   Syngenta, Jealott's Hill International Research Centre, Bracknell, UK
    *Corresponding author. Tel: +44 020 8383 8218; E-mail: andre.brown@imperial.ac.uk
    †These authors contributed equally to this work

Bivort, 2018). From previous symptomology work, it is clear that detailed postural information can be useful for resolving mode of action (Sluder *et al*, 2012; Salgado, 2017). We chose *C. elegans* as our model system because it is small and compatible with multiwell plates and automated liquid handling. It is sensitive to anthelmintics and insecticides and has played an important role in mode-of-action discovery in the past (Brenner, 1974; Sluder *et al*, 2012; Buckingham *et al*, 2014; Burns *et al*, 2015; Hahnel *et al*, 2020).

To combine the benefits of high throughput and high resolution, we used megapixel camera arrays to record the behavioral responses of worms to a library of 110 compounds covering 22 distinct modes of action. We simultaneously recorded all of the wells of 96-well plates with sufficient resolution to extract the pose of each animal and a high-dimensional behavioral fingerprint that captures aspects of posture, motion, and path. We show that worms have diverse dose-dependent behavioral responses to insecticides and anthelmintics and develop a machine learning approach that shares information across replicates and doses to accurately predict the mode of action of previously unseen test compounds. Furthermore, we show that a novelty detection algorithm can provide an indication that a compound belongs to a mode of action not seen in the training set. This novelty score can be used as a measure of confidence in the class prediction, suggesting a way to prioritize compounds with potentially novel modes of action early in the development process. These results demonstrate that high-throughput phenotyping in *C. elegans* is a promising approach for assisting target deconvolution in anthelmintic and insecticide discovery. Finally, we show that our prediction accuracy might be limited by uncertainties in the class definition rather than noise or phenotypic dimensionality. Specifically, we show that we can classify compounds even within a mode-of-action class, suggesting that there are limitations in our knowledge of the relevant pharmacology rather than limitations in our ability to reproducibly detect compound-induced phenotypes.

## Results

### Insecticides affect phenotypes in multiple behavioral dimensions

We assembled a library of 110 insecticides and anthelmintics with diverse targets to sample a range of modes of action used medically and commercially (see Dataset EV1 for full list). The modes of action represented in the library cover 70% of the market of insecticides used in the field (Sparks & Nauen, 2015) and several important classes of anthelmintics used in veterinary and human medicine (Nixon *et al*, 2020). To quantify the effects of the compounds on behavior, we recorded worms using megapixel camera arrays that simultaneously image all of the wells of 96-well plates (Fig 1A). We recorded at least 10 replicates at three doses for each compound with enough resolution to extract high-dimensional behavioral fingerprints following segmentation, pose estimation, and tracking (Fig 1A). The behavioral fingerprints are vectors of posture and motion features that are subdivided by body segment and motion state including "midbody curvature during forward crawling" or "angular velocity of the head with respect to the tail while the worm is paused". We have previously shown that similar features can detect even subtle behavioral differences that can be difficult to detect by eye (Yemini *et al*, 2013) and that the combined feature set has sufficient dimensionality to accurately classify worms with diverse behavioral differences caused by genetic variation and optogenetic perturbation (Javer *et al*, 2018a, 2018b).

As expected, several compound classes have strong visible effects on *C. elegans* behavior including the glutamate-gated chloride channel activator emamectin benzoate, the spiroindoline vesicular acetylcholine transporter inhibitor SY1713, and the serotonin receptor antagonist mianserin. All three compounds at specific doses can be distinguished from DMSO controls and from each other in a simple two-dimensional space defined by speed and body curvature (Fig 1B). The large differences in curvature and in motion caused by some compounds are observable by eye, as shown in the inset images and in Fig 1C. However, not all compounds are well separated in these two dimensions; the gray points in Fig 1B show the dose means and standard deviation of all the compound doses in the speed/tail curvature space, which largely overlap. Some of the screened compounds might not have detectable effects on *C. elegans* and therefore cannot be used for phenotypic mode-of-action prediction. To find the compounds with no effect, we compared the behavioral fingerprints of treated worms with DMSO controls using univariate statistical tests for each feature and correcting for multiple comparisons with the Benjamini–Yekutieli procedure (Benjamini & Yekutieli, 2001). To account for random day-to-day variation in the experiments, we used a linear mixed model for these statistical tests, where the fixed effect is the drug dose and the day of the experiment is added as a random effect. The number of features that are significantly different at a false discovery rate of 1% between the

---

**Figure 1. Insecticides affect phenotypes in multiple behavioral dimensions.**

A We image an entire 96-well plate with a megapixel camera array with enough resolution and high enough frame rate to track, segment, and estimate the posture of *C. elegans* over time.

B All compound doses in the speed/tail curvature space, with points and lines showing the mean and standard deviation of biological dose replicates. On average, 12 biological replicates were collected per compound dose, together with 601 DMSO replicates, across at least 3 different tracking days for each condition. Several compounds, including the serotonin receptor antagonist mianserin (blue), glutamate-gated chloride channel activator emamectin benzoate (purple), and vesicular acetylcholine transporter inhibitor SY1713 (red), have a strong effect on the worms' behavioral phenotype. They can be distinguished from the DMSO control (black) and from each other based on speed and tail curvature alone. Not all compounds are well separated in these two dimensions (gray points). Inset images are samples that show postural differences.

C Sample worm skeletons over time show the effect of the compounds highlighted in (B) on motion.

D Number of features significantly different from the DMSO control at a false discovery rate of 1% for each compound, grouped by mode of action. The pre-stimulus, blue light stimulus, and post-stimulus data are shown separately (a total of 3,020 features are tested for each assay period). The percentage of significantly different features is highest for the blue light stimulus recording.

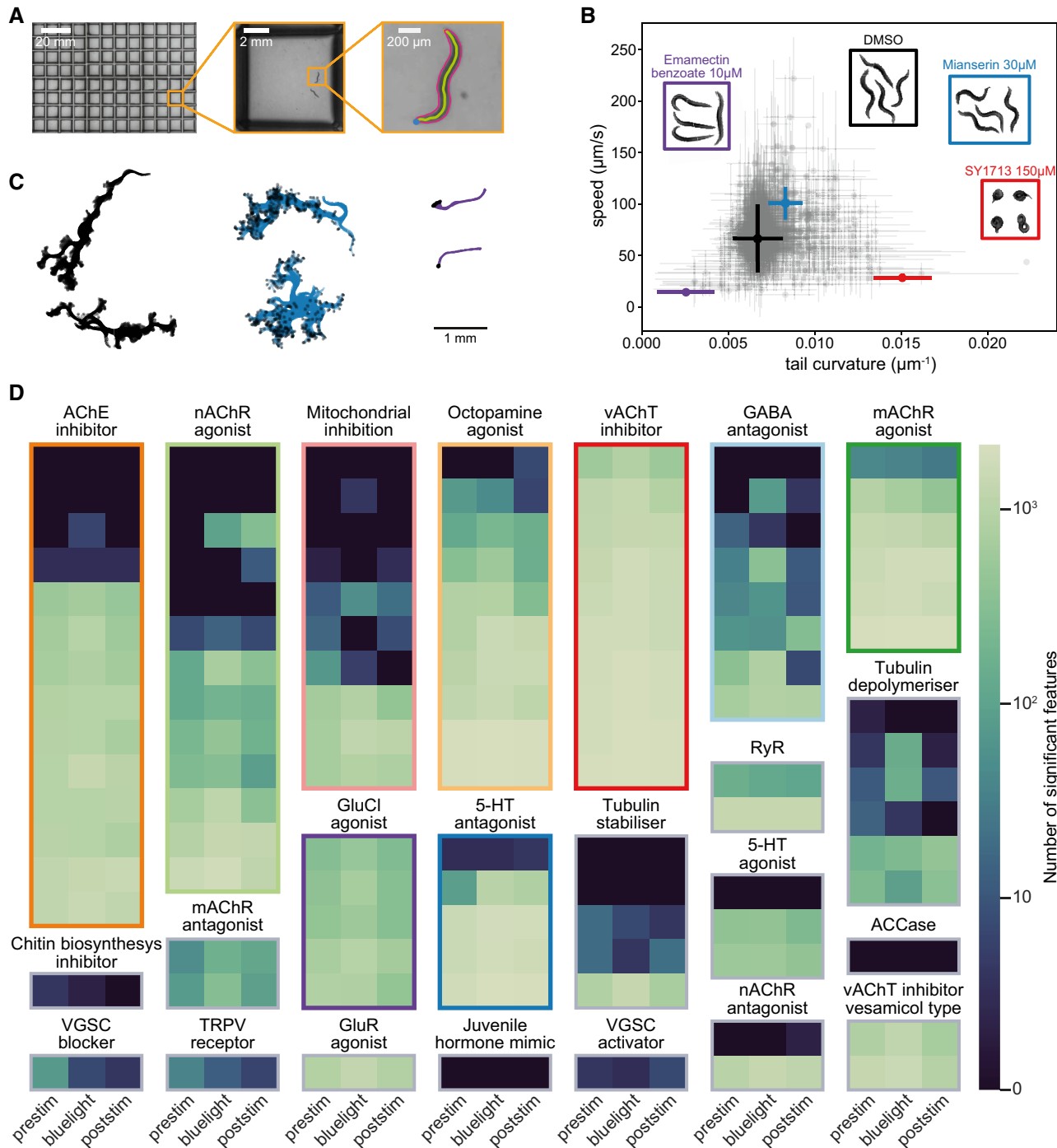

**Figure 1.**

behavior of worms treated with each compound and the DMSO controls is summarized in a heat map (Fig 1D).

To further increase the dimensionality of the behavioral phenotypes, we included a blue light stimulation protocol. Each tracking experiment is divided into three parts: (i) a 5 min pre-stimulus recording, (ii) a 6-min stimulus recording with three 10-s blue light pulses starting at 60, 160, and 260 s, and (iii) a 5 min post-stimulus recording. Blue light is aversive to *C. elegans* (Edwards *et al*, 2008) and so it can help to distinguish between animals that are simply pausing and

those that are not able to move (Churgin *et al*, 2017). Behavioral differences are observed in each assay period, but the stimulus period shows the most differences (Fig 1D). Even within mode-of-action classes, compound potency can be highly variable. The largest potency difference is observed for the octopamine agonists where amidine affects 0.08 % of features and oxazoline affects 75 % of features. Overall, 86 % of compounds have a detectable effect on behavior in at least one feature. The 17 compounds that showed no detectable effect in any stimulus period were not included in subsequent analysis.

## Compounds with the same mode of action have similar effects on behavior

Having established that *C. elegans* shows diverse behavioral responses to insecticides and anthelmintics, we next sought to determine to what extent the responses are mode-of-action-specific. For the initial clustering, we used 256 features from each blue light condition. These features were selected for their usefulness in classifying mutant worms in a previous paper (Javer *et al*, 2018b). For all clustering and classification tasks, we first z-normalize each feature to put them on a common scale and to prevent arbitrary choices of

units from impacting the analysis. We used hierarchical clustering to visualize the relationships between the behavioral responses to different compounds at different doses (Fig 2A). Each row of the heat map is the average of all of the replicates of a given compound at a specific dose. We also included the averages of six subsets of the DMSO replicates randomly partitioned across tracking days as control points. Several of the compound classes show clear clustering, including the AChE inhibitors, vAChT inhibitors, GluCl agonists, and mAChR agonists. The DMSO averages also cluster closely together. The degree of mode-of-action clustering is greater than expected by chance, which can be seen in a plot of the cluster purity

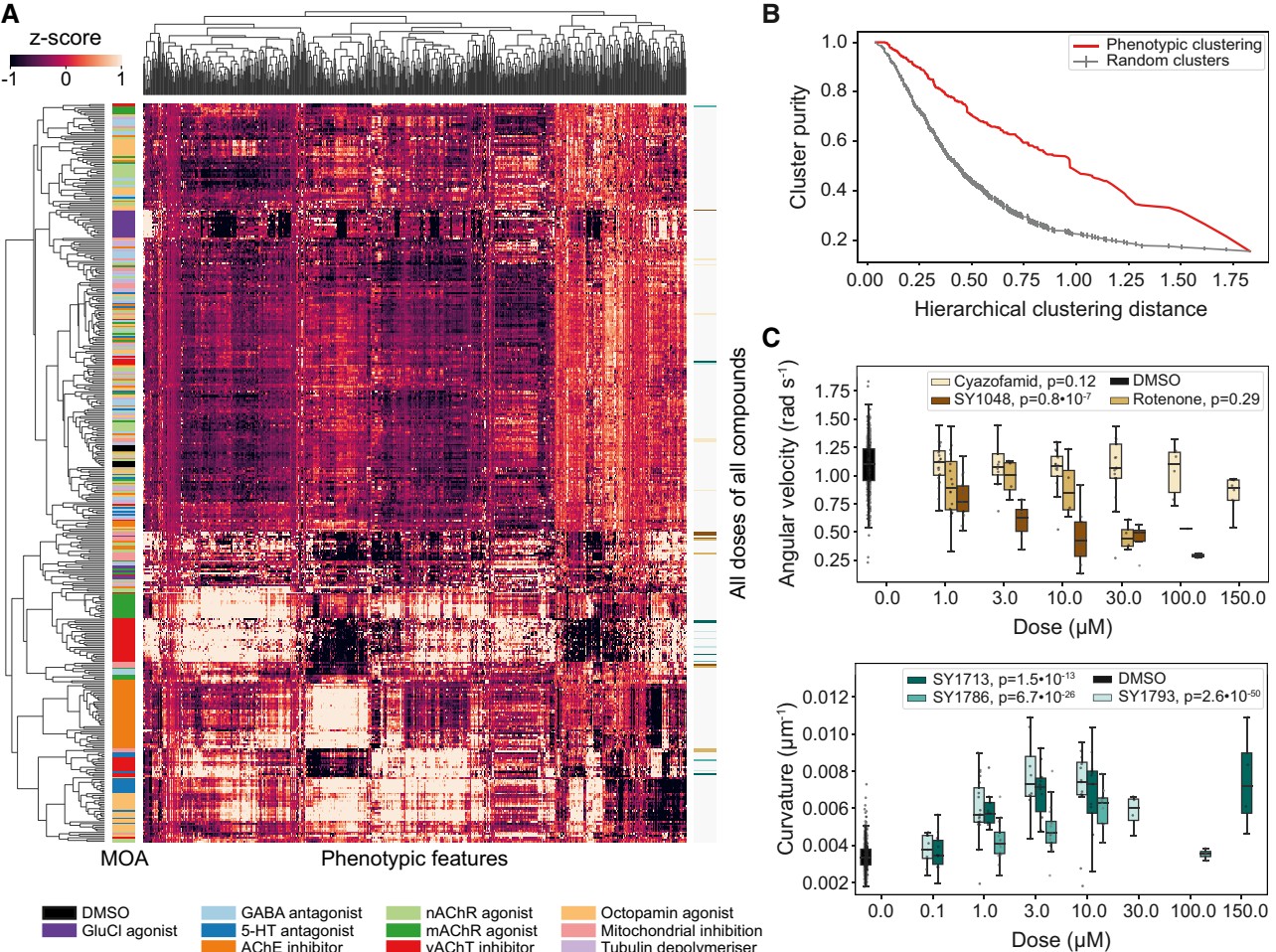

**Figure 2. Clustering and dose response of behavioral fingerprints.**

A   Hierarchical clustering of behavioral fingerprints highlights structure in the responses to different compounds. Each row of the heat map represents the mean dose fingerprint of a specific compound described by 256 pre-selected features from each blue light condition. Clear clusters can be observed for some compound classes, e.g., AChE inhibitors, vAChT inhibitors, GluCl agonists, and mAChR agonists. Low doses and low potency compounds from different classes cluster together around the DMSO averages at the center-top part of the heat map.

B   Cluster purity as a function of the hierarchical cluster distance shows that the degree of mode-of-action clustering (red) is greater than expected by chance for random clusters (gray).

C   Compounds in the same class can have different dose–response curves. (upper) The three mitochondrial inhibitors cyazofamid, rotenone, and SY1048 all decrease angular velocity, but the concentration at which their effect is measurable is not conserved across compounds. (lower) Different spiroindolines affect body curvature differently. SY1786's dose–response curve is non-monotonic. The central band and box limits show the median and quartiles of the distribution of the biological replicates for each compound dose (on average 12 wells per dose and 601 DMSO wells), while the whiskers extend to 1.5 IQRs beyond the lower and upper quartile. The P-values reported in the legend represent the significance of the drug dose effect and were estimated using linear mixed models with tracking day as random effect and drug dose as fixed effect. The positions of these compounds in the heat map in (B) are marked using the color bar on the right side of the heat map.

observed in the data compared with random clustering (Fig 2B). However, the distance between compounds that share the same mode of action can be large, even for classes that cluster well overall, in part because behavioral fingerprints change with dose. It is also not always possible to align feature vectors using doses because compounds can have very different potencies: A low dose for one compound could be a high dose for another. Furthermore, the compound concentration inside the worm is likely to be much lower than the concentration in the media because of *C. elegans*' considerable xenobiotic defenses (Hartman *et al,* 2021) and the degree of uptake will also vary across compounds even within a mode-of-action class. For this reason, the center-top part of the heat map in Fig 2A is populated with low doses and compounds that either have a low potency or low uptake in the worm, which do not form distinct clusters based on their mode of action, but are rather clustered around the DMSO averages.

These effects can be seen in dose–response plots for individual features. The three mitochondrial inhibitors in Fig 2C all decrease angular velocity, but they do it at different doses. At 3 μM, only SY1048 has a strong effect, while at 30 μM, rotenone has a similarly strong effect. Clustering based on angular velocity would lead to qualitatively different conclusions about nearest neighbors at these different doses. For the spiroindolines, similar differences in dose–response are observed for body curvature with the added difference that the effect of SY1786 is non-monotonic and returns to baseline at high doses. These non-monotonic effects can be due to compounds precipitating from solution at high doses or due to intrinsically complex compound effects such as a compound that causes an increase in speed at low doses but is lethal at high doses. Regardless of the cause, complex dose–response curves present challenges for mode-of-action prediction since supervised machine learning algorithms rely on differences in feature distributions to learn decision boundaries and dose–response effects spread out the distributions and increase the overlap between classes.

## Combining classifiers by voting enables mode-of-action prediction

The behavioral fingerprints of compounds with the same mode of action have the same direction in the phenotypic space and can be used for classification in mode-of-action classes. For the classification task, we need a minimum number of compounds per class to get an accurate representation of the class distribution. Out of the compounds with detectable effects in *C. elegans*, we choose only the classes with at least five compounds (10 classes with 76 compounds). We take advantage of the fact that several replicates are recorded per condition and resample with replacement from the multiple replicates for each dose to create a set of average behavioral fingerprints. This effectively smooths the data reducing the effect of outliers. At the same time, it provides a simple method for balancing classes before classifier training. For classes with fewer compounds, we resample more times so that each class contains the same number of points (see Fig EV1). To partially mitigate the effect of compound potency, we then normalize each behavioral fingerprint to unit magnitude. This normalization is done row-wise on each sample in contrast to the z-normalization described above which is done column-wise on each feature. Rescaling in this way brings compounds with similar effect profiles but different potencies closer together in feature space

(Figs 3A and EV2), but because of nonlinearities in the dose–response profiles, the overlap is not perfect even after rescaling.

Predictions must be combined across doses and replicates to make a single prediction for the mode of action of a given compound. Inspired by an analogy with the multi-sensor fusion problem (Singh *et al,* 2019), we use a voting procedure to make a final prediction. However, in contrast to multi-sensor fusion, we cannot train different classifiers for each dose because 1 μM for one compound is not equivalent to 1 μM for another compound. Instead, we train a single classifier for all doses and make predictions for each data point. Each data point contributes a vote for a compound's class and the class with the most votes wins.

We split our data into a training/tuning set consisting of 60 compounds and a hold-out/reporting dataset consisting of 16 compounds containing at least one compound for each mode-of-action class. For mode-of-action prediction, we started with the full set of features output by Tierpsy (3,020 per blue light condition for 9,060 in total) and used the training set to determine an appropriate classifier, select features, and tune hyperparameters using cross-validation. We achieved the highest cross-validation accuracy with 1,024 features selected using recursive feature elimination with a logistic regression estimator. The hyperparameters of the classifier were also tuned using cross-validation, and the best version of the estimator was multinomial logistic regression with l2 regularization and penalty parameter $C = 10$. Using a regularized linear classifier helps to control overfitting in this high-dimensional feature space, which boosts the cross-validation accuracy. The confusion matrix from cross-validation using the best performing feature set and classifier is shown in Fig 3. To determine whether the classifier could generalize to unseen compounds, we applied it to the test data without further tuning. The classifier predicted the correct mode of action for the unseen compounds 88% of the time (Fig 3C). We generated a null model by partitioning the DMSO data randomly across tracking days to 10 classes. Following the same steps as for the compound-treated data, we obtained a maximum cross-validation accuracy of 10% using the training set, while the prediction accuracy in the test set was 12.5%.

In addition to the 10 modes of action that were represented by at least five compounds in our dataset, we had 17 compounds with a detectable effect on *C. elegans* that belonged to 11 sparsely populated mode-of-action classes. We used these additional compounds to simulate another use case for our approach: detecting screening hits that represent potentially novel modes of action that do not fall into known classes. We use the term "novel test set" to describe these compounds, since their modes of actions are unknown (novel) to the trained classifier. Using a novelty detection algorithm (preprint: Vinokurov & Weinshall, 2016) with some modifications, we assigned a novelty score to each of the test and novel test compounds based on their affinity to each of the existing classes. To obtain the novelty score, we use an ensemble of support vector machine (SVM) classifiers that flag novel compounds based on the confidence values of the main multinomial logistic regression classifier used for the predictions of known classes. The ensemble of SVM classifiers is trained using partitions of the training set into presumed-known and presumed-unknown classes. The novelty score is defined as the weighted average of the output of this ensemble. Most of the novel compounds were assigned novelty scores above 0.8 (Fig 3D). Several of the non-novel compounds—those

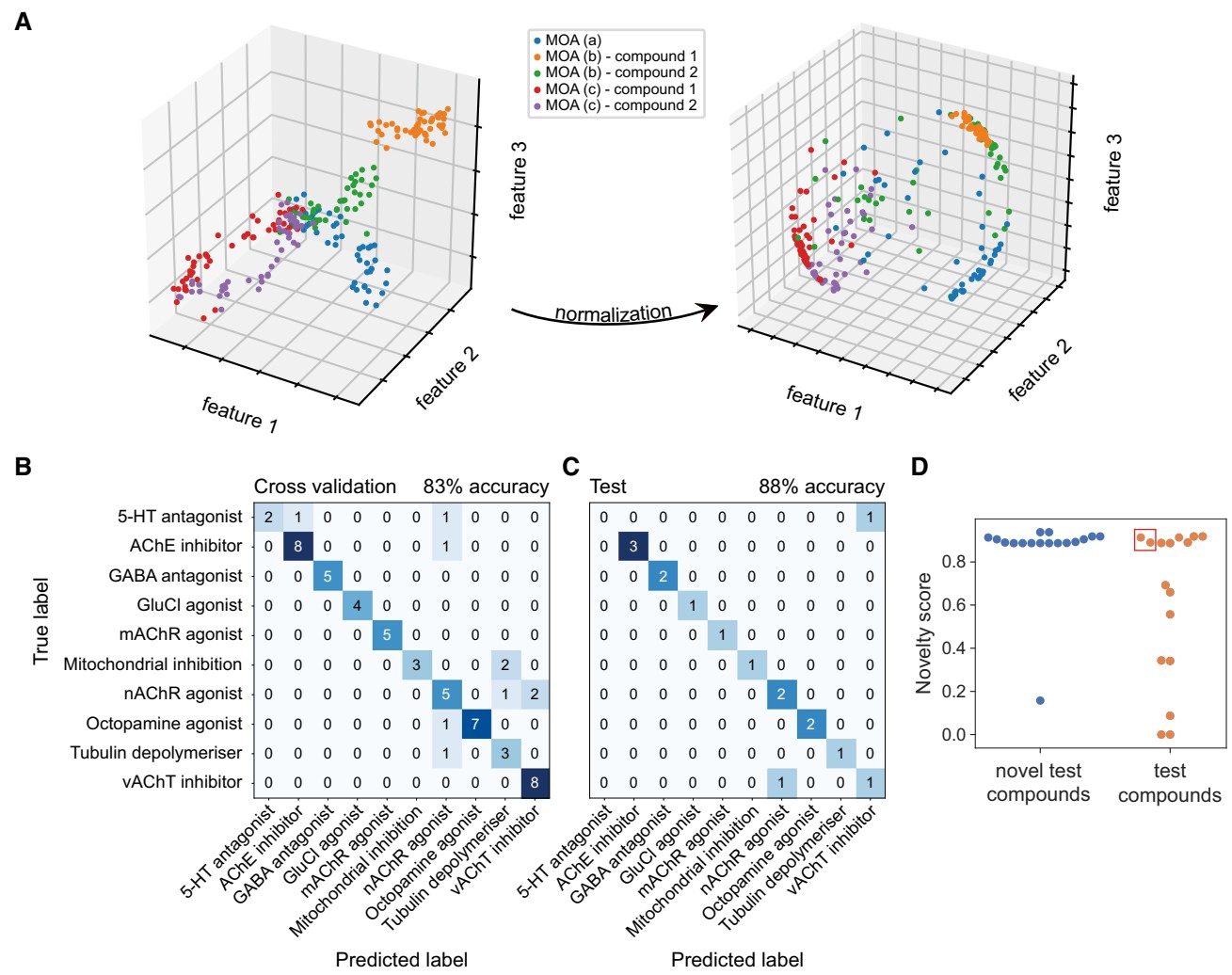

**Figure 3. Classifiers trained on behavioral fingerprints can predict the mode of action of unseen test compounds.**

A  Toy data illustrating the potential benefit of normalization in correcting for potency differences within mode-of-action classes. Following normalization, each behavioral fingerprint exists on a hypersphere in the phenotype space regardless of effect size in the original space. Nonlinear dose–response curves will not collapse perfectly following normalization, which is a linear transformation.

B  The confusion matrix obtained through cross-validation for the best performing feature set (1,024 features) and logistic regression classifier following feature selection and hyperparameter tuning on the training data.

C  The confusion matrix for the classifier trained in (B) applied to previously unseen test compounds without any further tuning.

D  The novelty score assigned to novel test compounds with a mode of action not seen during training compared to the novelty score of compounds from the test set in (C). Novel compounds tend to have higher novelty scores than compounds from previously seen modes of action. The non-novel compounds with high novelty scores include the two incorrectly classified test compounds (in red box).

that come from a class that is present in the training data—have high novelty scores, but this includes the two test compounds that were incorrectly classified. In this case, the high novelty score correctly indicates low confidence in the prediction of the classifier. To explore the origin of the high novelty score for the incorrectly classified compounds, we looked for differences between the effects of compounds within a class.

**Mode-of-action deconvolution within classes**

Although compounds are categorized into broad mode-of-action classes, most compounds will have some degree of off-target

engagement. If the off-target effects are different for compounds within a mode-of-action class or if the compounds have differences in pharmacokinetics, they may lead to different phenotypes. In this case, it may be possible to use behavioral fingerprinting to further deconvolve mode-of-action classes revealing hidden compound heterogeneity. To test for phenotypic differences within mode-of-action classes, we trained a classifier to distinguish the replicates from each compound within a class from the replicates of the other compounds in the class. We then used cross-validation accuracy to quantify the distinguishability of the compounds within a class. We hypothesized that for classes without mechanistic sub-classes, the classifier would perform similarly to random guessing. In contrast,

if compounds with the same mode of action had different off-target profiles or different pharmacokinetics, the classifier would be able to reliably distinguish individual compounds or subsets of compounds with the same broad mode of action.

In all classes, the compound-level classifiers performed better than random guessing, in some cases by a large margin. One of the incorrectly classified compounds in the test set was ritanserin, which was also assigned a high novelty score. The within-class classifier shows that it is indeed clearly distinguishable from the other 5-HT receptor antagonists (Fig 4A). Although ritanserin is known to be a 5-HT receptor antagonist, it is also known to affect multiple other targets. In addition to detecting outlying compounds, the deviations from random guessing revealed substructures within the classes. For example, one group contains the two antidepressants, which are nearest neighbors in terms of structural similarity (atom pair Tanimoto coefficient of 0.44 between the antidepressants compared with a Tanimoto coefficient of $0.20 \pm 0.09$ (mean $\pm$ SD) for the other pairwise comparisons within the class). As with ritanserin, the other four compounds in this class have known polypharmacology, which could be driving their clustering. Another class

with interesting substructure is that consisting of mitochondrial inhibitors, which also separates into distinguishable groups (Fig 4B). In this case, the phenotypically distinct groups separate the complex I inhibitors from the complex II and complex III inhibitors, which appear phenotypically more similar. The mectins we tested are structurally similar and are known to share the same binding site, suggesting they would be difficult to separate into subgroups. Consistent with this expectation, the compound-level mectin classifier performs only slightly better than chance (Fig EV3).

## Discussion

We have shown that worms have diverse responses to insecticides and anthelmintics and that behavioral fingerprints can be used to cluster compounds with similar modes of action. With appropriate normalization and by combining information across doses and replicates through voting, we can also accurately predict the mode of action of previously unseen compounds despite differences in compound potency and uptake into the worm. Improving

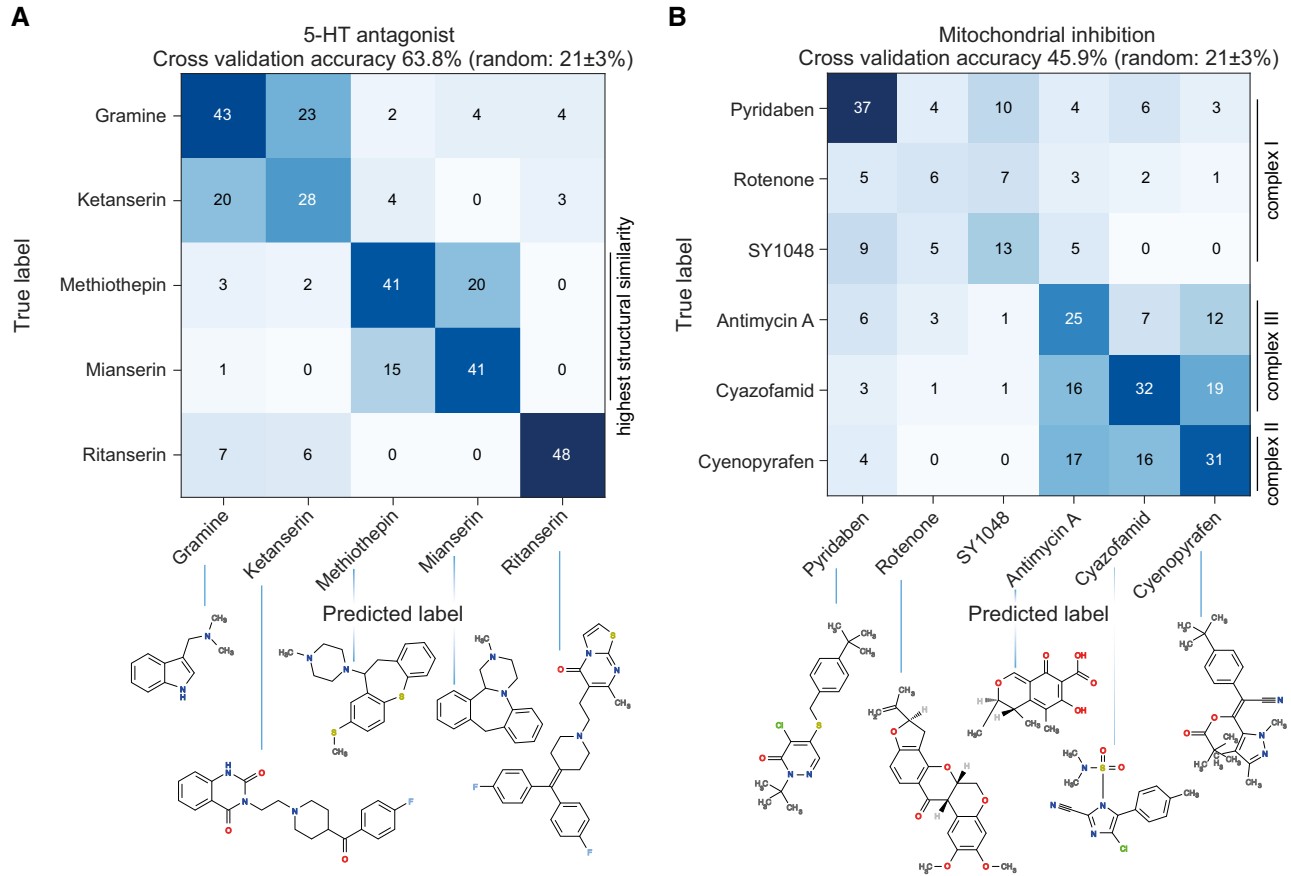

**Figure 4. Mode of action can be resolved within compound classes.**

A  The confusion matrix showing cross-validation performance of a classifier trained to distinguish serotonin receptor antagonists from each other. Ritanserin is distinguishable from all other compounds, and the two structurally similar antidepressants (mianserin and methiothepin) are somewhat mutually confused by the classifier but are distinguishable from the non-antidepressants.

B  The confusion matrix for the mitochondrial inhibitors also shows some substructure: complex II and III inhibitors (cyazofamid, antimycin, cyenopyrafen) are phenotypically similar, and distinct from the complex I inhibitors.

compound delivery through non-specific means such as using cuticle disrupting mutants (Xiong *et al*, 2017) could equalize the internal concentrations of compounds and reduce this source of variation in the data. Improved delivery may increase the number of compounds with detectable effects on behavior and improve the accuracy of mode-of-action predictions.

Given the wide variety of modes of action included in this study, it might be expected that the dimensionality of worm behavior would be limiting. In other words, it could have been the case that worm behavior can only vary in a small number of ways and that this would limit the number of distinct classes that are distinguishable using tracking data alone. On the contrary, we found that in addition to predicting mode of action across ten classes, we could often classify at the level of individual compounds and that the within-mode-of-action confusion matrices revealed subgroups that related to finer-scale mechanistic divisions. This finding suggests there is sufficient dimensionality in worm behavioral responses to detect mode of action and more detailed pharmacological differences such as the spectrum of a compound's off-target effects or differences in target engagement. These results belie worms' superficial simplicity but are consistent with genetic findings that mutations can lead to multiple types of uncoordination (Brenner, 1974) and with tracking results, suggesting that even wild-type spontaneous locomotion is surprisingly complex (Schwarz *et al*, 2015; Gomez-Marin *et al*, 2016).

The most obvious limitation of our method is that not all compounds affect the N2 strain of *C. elegans*. Only a small minority of the compounds we assayed had no detectable effect, but there will be entire classes of compounds that are not expected to have an effect on *C. elegans* because their targets are not conserved, such as the pyrethroid insecticides that target voltage-gated sodium channels (Vijverberg & van den Bercken, 1990). Expanding the range of organisms included in the training data is one way to address this limitation. There are methods for deriving multidimensional behavioral fingerprints that incorporate postural information from flies (Berman *et al*, 2014) and zebrafish larvae (Marques *et al*, 2018), and both organisms are compatible with high-throughput screening. Because our approach already involves the fusion of data from multiple samples, additional species-level classifiers could be seamlessly incorporated into the voting procedure to arrive at a final prediction. Results from ensemble learning in diverse fields suggest a further benefit of a multi-species approach: Increasing the votes from independent classifiers should increase classifier accuracy if the predictions from different species are partially uncorrelated (Kittler *et al*, 1998). The sample principle applies beyond behavioral phenotypes. The results of automated symptomology can also be combined—within the same analysis framework—with data from non-animal species including bacteria and fungi as well as other assays commonly used in mode-of-action identification including genetic and biochemical assays.

Strains of *C. elegans* that have been recently isolated from the wild are readily available (Cook *et al*, 2016) and have given insight into anthelmintic resistance (Ghosh *et al*, 2012). Using wild-isolated strains of *C. elegans* would increase the diversity of compound responses without having to modify the experimental protocols for screening or downstream analysis. Moving beyond *C. elegans* to other nematode species, including parasitic nematodes, could provide further independent phenotypes for improving mode-of-action prediction. However, particularly for animal parasites, different morphologies might require alternative analysis approaches (Marcellino *et al*, 2012; Buckingham *et al*, 2014; Partridge *et al*, 2018).

Phenotypes derived from *in vitro* microscopy of cells in culture have also been extensively studied for mode-of-action prediction (Perlman, 2004; Ljosa *et al*, 2013; Caicedo *et al*, 2017). Human cells in culture would provide an additional phylogenetically distinct species to combine with invertebrate screening. Human cell responses may also give an indication of mammalian toxicity (Kleinstreuer *et al*, 2014). However, there are drawbacks with respect to insecticide and anthelmintic discovery. The first, which is shared with zebrafish, is that the goal of these compounds is to affect invertebrates without affecting vertebrates so there may be a lower response rate than for pharmaceutical compounds. Cultured insect cells (Douris *et al*, 2006) could be used in similar assays as human cells but may provide more relevant information for insecticide mode of action. The second drawback, which applies to most culture systems, is that some compounds act on super-cellular structures. For example, acetylcholinesterase inhibitors act by causing a buildup of acetylcholine at synapses or neuromuscular junctions. Whole-animal phenotypic screening is therefore likely to continue to play a role in phenotypic screening of neuroactive compounds both for discovery and for mode-of-action prediction.

Symptomology is an important technique for mode-of-action determination in insecticide discovery. Our method would therefore fit straightforwardly into existing discovery pipelines. Since phenotypic screening in pest species is the primary means of lead identification, an intriguing possibility would be to adapt our method to primary screening data so that mode-of-action prediction is not only available for selected hits but can already be included at the earliest stages of decision making. Given advances in computer vision and pose estimation (Mathis *et al*, 2018; Graving *et al*, 2019; Pereira *et al*, 2019), it should be possible to track pest species in complex media such as leaf sections with existing technology. Improved phenotyping in primary screens might also reduce false-negative rates by picking up compounds with unique phenotypic effects that are too weak to register as hits using current metrics but might provide useful starting points for optimization.

# Materials and Methods

### Reagents and Tools table

| Reagent/Resource | Reference or Source | Identifier or catalog number |
| --- | --- | --- |
| **Experimental models** | | |
| N2 (*C. elegans*) | Caenorhabditis Genetics Center (CGC) | N/A |

**Reagents and Tools table** (continued)

| Reagent/Resource | Reference or Source | Identifier or catalog number |
|---|---|---|
| OP50 (*E. coli*) | Caenorhabditis Genetics Center (CGC) | N/A |
| **Chemicals, enzymes and other reagents** | | |
| Bactopetone (NGM) | ThermoFisher | 211820 |
| Bio-agar (NGM) | Biogene | 400-050 |
| CaCl2 (NGM) | Sigma | C3881-1KG |
| Cholestrerol (NGM) | Sigma | C1145-250MG |
| DMSO | Sigma-Aldrich | 276855-100ML |
| Ethonol (NGM) | VWR | 20823.362 |
| $KH_2PO_4$ (M9) | Sigma-Aldrich | P0662-500G-M |
| $MgSO_4$ (M9) | Fisher | M/1050/53 |
| NaCl (M9/NGM) | Sigma- Aldrich | 71376-1KG |
| $NaHPO_4$ (M9) | VWR | 28040.291 |
| Sodium Hydroxide 1 M (Bleach) | Merck Millipore | 1091371000 |
| Sodium hypochlorite, 5% Chlorine (Bleach) | Fisher Scientific | #419550010 |
| **Software** | | |
| tierpsy-tracker ver 1.5.2-alpha+3c2c254 | https://github.com/Tierpsy/tierpsy-tracker | |
| tierpsy-tools-python ver 0.1 | https://github.com/Tierpsy/tierpsy-tools-python | |
| Well-annotator | https://github.com/Tierpsy/WellAnnotator | |
| **Other** | | |
| Whatman UNIPLATE 96-Well Clear Microplates | VWR international Ltd | WHAT7701-1651 |
| Whatman Clear Universal Microplate Lid | VWR international Ltd | WHAT-77041001 |
| VIAFILL Bulk Reagent Dipenser | INTEGRA | 5600 |
| VIAFLO 96, 24 and 96 channel handheld pipette | INTEGRA | 6001 |
| acA4024-29um - Basler ace (camera) | Basler | 107404 |
| HF3520-12 M 1" (lens) | FUJIFILM | HF3520-12 M |
| Hydra Imaging Rig | LoopBio GMBH | N/A |

## Methods and Protocols

### Worm husbandry

The N2 Bristol *C. elegans* strain was obtained from the CGC (*Caenorhabditis* Genetics Center) and cultured on Nematode Growth Medium at 20°C and fed with *E. coli* (OP50) following standard procedures (Brenner, 1974).

### Worm preparation

Synchronized populations of young adult worms for imaging were cultured by bleaching unsynchronized gravid adults, and allowing refed L1 diapause progeny to develop for two and a half days (a version of the protocol is maintained at protocols.io https://dx.doi.org/10.17504/protocols.io.2bzgap6). On the day of imaging, young adults were washed in M9 (a version of the protocol is maintained at protocols.io https://dx.doi.org/10.17504/protocols.io.bfqbjmsn) and transferred to the prepared drug plates (2 worms per well for the first round of imaging in December 2019 and 3 worms per well in the second round in January 2020) via the COPAS 500 Flow Pilot (a version of the protocol is maintained at protocols.io https://dx.doi.org/10.17504/protocols.io.bfc9jiz6) and returned to a 20°C incubator for 3.5 h. Plates were then transferred onto the multi-camera

tracker for another 30 min to habituate prior to imaging so that the total drug exposure time was 4 h.

### Plate preparation

Low peptone (0.013%) nematode growth medium (a version of the protocol is maintained at protocols.io https://dx.doi.org/10.17504/protocols.io.2rcgd2w) was prepared as follows: 20 g agar (Difco), 0.13 g bactopeptone, and 3 g NaCl were dissolved in 975 ml of Milli-Q water. After autoclaving, 1 ml of 10 mg/ml cholesterol was added along with 1 ml of 1 M $CaCl_2$, 1 ml 1 M $MgSO_4$, and 25 ml of 1 M $KPO_4$ buffer (pH 6.0). Molten agar was cooled to 50–60°C, and 200 µl was dispensed into each well of 96-square well plates (WHAT-7701651) using an Integra VIAFILL (a version of the protocol is maintained at protocols.io https://dx.doi.org/10.17504/protocols.io.bmxbk7in). Poured plates were stored agar side up at 4°C until required.

Prior to applying compounds, plates were placed without lids in a drying cabinet to lose 3–5% weight by volume and then stored with lids (WHAT-77041001) at room temperature until required.

### Compound preparation

Compounds were prepared for screening by dissolving in DMSO at 1,000× their final imaging plate concentration (so that final

concentration of DMSO in imaging plates was 0.01%). The results presented here are pooled from two rounds of experiments that used two different methods of adding compounds.

For the first round (December 2019), the library was stored in 56 "master" 96-well plates in which eight replicates of each compound solution were in a single column of a 96-well plate and lower concentrations were made by serial dilution of the highest concentration using DMSO. There were up to six doses per drug and one column was reserved for DMSO and one column for no compound controls on each master plate, which were stored at −20°C. The day prior to imaging, "source" plates were prepared using an Integra VIAFLO 96-channel pipette by adding 7 µl water and 0.5 µl 1,000× drug and mixing by pipetting up and down. 5 µl water was added to the surface of each well of an imaging plate to facilitate compound transfer and prevent agar damage from pipette contact. Then, 3 µl of compound was transferred from the source plates to the destination imaging plates using an Opentrons liquid handling robot, which randomly shuffled the column order during the transfer (Fig EV4).

For the second round (January 2020), "master" 96-well plates were filled sequentially with three doses of each drug (made by serial dilution of the highest concentration using DMSO) so that the entire library fitted into three and a half 96-well plates. Three sets of shuffled "library" plates were created with randomized column orders using an Opentrons robot so that three replicates per drug per dose fitted into 10.5 96-well plates. All plates were stored at −20°C. The day prior to imaging, "source" plates were prepared using the VIAFLO by transferring 1.4 µl of 1,000× drug from the shuffled "library" plates into 96-well (PCR) plates filled with 19 µl of water and pipetted up and down to mix. 5 µl of water was added to the imaging plates to facilitate compound transfer, and then, 3 µl of the pre-diluted compounds was then transferred to imaging plates using the VIAFLO.

After the drug had absorbed into the agar, imaging plates were seeded with 5 µl OP50 diluted 1:10 in M9 solution using an Integra VIAFILL and left overnight at room temperature in the dark. Full protocols are available at https://doi.org/10.17504/protocols.io.9vqh65w and https://doi.org/10.17504/protocols.io.bn5zmg76.

### Image acquisition

Image acquisition was performed on five custom-built tracking rigs (LoopBio GMBH, Vienna). Each rig features six Basler acA4024 cameras (Basler AG, Ahrensburg, Germany) arranged in a 3 × 2 array and equipped with Fujinon HF3520-12 M lenses (Fujifilm Holding Corporation, Tokyo, Japan) and long-pass filters (Schneider-Kreuznach IF 092; Schneider-Kreuznach, Germany, and MidOpt LP610; Midwest Optical Systems Inc., Palatine, IL, USA). This allows us to simultaneously image all the wells of a 96-well plate at up to 30 frames/s and with 12.4 µm/px resolution. Blue light stimulus was provided using four Luminus CBT-90 TE light-emitting diodes with a peak wavelength of 456 nm and peak radiometric flux of 10.3 W each.

To avoid any light avoidance response during the off-stimulus imaging periods, we used a near-infrared (850 nm) LED panel with two collimation filters (3 M; St. Paul, MN, USA) to provide uniform bright-field illumination.

Each imaging rig is connected to two Dell workstations with an Intel i7 CPU (Intel Corporation, Santa Clara, CA, USA), 16 GB of RAM, an Nvidia Quadro P2000 (Nvidia Corporation, Santa Clara, CA, USA), and running Ubuntu 18.04 LTS (Canonical Ltd., London, UK). Each machine handles the data acquisition of three cameras,

via as many USB3 connections. Loopbio's Motif software and API are used to control the image acquisition and light stimulation.

Each tracking experiment is divided into three parts, which are run in series by a script: (i) a 5 min pre-stimulus recording, (ii) a 6 min stimulus recording with three 10-s blue light pulses starting at 60, 160, and 260 s, and (iii) a 5 min post-stimulus recording.

### Image processing and quality control

Segmentation and skeletonization were performed using Tierpsy tracker (Javer et al, 2018a) (https://github.com/Tierpsy/tierpsy-tracker). Each video was manually checked using the Tierpsy tracker viewer, and any wells that had precipitation, excess unabsorbed liquid that led to swimming worms, or damaged agar were marked as bad and excluded from the analysis.

### Feature extraction and pre-processing

Python code to reproduce the analysis in the paper is available on GitHub (https://github.com/Tierpsy/moaclassification). We used Tierpsy tracker to obtain summarized behavioral features for each screened well (Javer et al, 2018b). These include morphological features such as length and width, postural features such as curvature, and features describing movement such as speed and angular velocity. A total of 3,020 features is obtained. We derived summarized features separately for the pre-stimulus period, for the period with blue light stimuli, and for the post-stimulus period resulting in a total of 9,060 features for each well.

In addition to the manual quality control described above, we used the filtering options in Tierpsy tracker to filter out tracked objects that have average width smaller than 20 µm or larger than 500 µm and average length smaller than 200 µm or larger than 2,000 µm. We also removed wells in which the number of tracked skeletons is smaller than 50 and wells for which more than 20% of the features could not be evaluated. Features that have more than 5% NaN values in the entire dataset are removed entirely. Any remaining NaN values are imputed to the mean of the given feature calculated across all wells. Compounds with very low effect compared with DMSO are also removed from the dataset. A compound is considered to have very low effect when none of the features shows significant differences across doses (including DMSO as dose 0) in a statistical test based on a linear mixed model with random intercept where the dose is the fixed effect and the day of the experiment is the random effect. The Benjamini–Yekutieli method (Benjamini & Yekutieli, 2001) with 1% false discovery rate was used to correct for multiple comparisons. Out of 110 compounds that were screened, 17 do not show a detectable effect in any of the features based on the univariate statistical tests and are therefore dropped from subsequent analysis. After cleaning up the dataset, we standardize the features (to mean 0 and standard deviation 1) to bring them on a common scale and prevent the unit differences from influencing the analysis.

For the classification task, we use a bootstrap method that simultaneously smooths the data and balances the classes. We bootstrap from the dose replicates of every compound (i.e., we resample with replacement until we get a sample 0.6 times the size of the initial sample) multiple times and each time we derive the mean feature values. In this way, we replace the drug dose replicates with bootstrapped averages, which smooth the data reducing the effect of outliers. This method also gives an easy way to balance the classes.

The number of bootstrapped averages per dose in the training set depends on how well-populated the mode-of-action class is. In the class with the most members (AChE inhibitors with 9 compounds), we get 20 bootstrap averages per drug dose. In the rest of the classes, the number of averages is proportional to the ratio between the max number of members in a class (9) and the number of members in the given class. Finally, to partially mitigate the effect of compound potency, we normalize each behavioral fingerprint to unit L2 norm. Rescaling in this way brings compounds with similar effect profiles across features but different potencies closer together in feature space.

### Train, test, novel split

For the classification task, we created a training set and a test set. We considered that we need at least 4 compounds per class in the training set for cross-validation, so classes with less than 5 compounds were not included. As a result, we used ten classes for the classification task with a total of 76 compounds. The partitioning in training set and test set was done using stratified split as implemented in scikit-learn (Pedregosa *et al*, 2011) with 20% of the compounds of each class included in the test set. Using this method, 60 compounds were assigned to the training set, while 16 compounds spanning all ten classes were assigned to the test set. The compounds of the sparsely populated classes that are not included in the classification task were considered novel test compounds and were used for novelty detection. Eleven classes with a total of 17 compounds are included in the novel test set. The training set was used for feature selection and hyperparameter selection. The test set was used to test the classification accuracy of the selected trained model. Both the test set and the novel test set were used to test the novelty detection method. The pre-processing and splitting of the data are described in a flow chart in Fig EV5.

### Hierarchical clustering

To investigate how well the compounds of the same mode-of-action cluster together in an unsupervised way, we use hierarchical clustering. For this task, we use the 256 features in the tierpsy_256 feature set (Javer *et al*, 2018b) from every blue light condition. We need a reduced feature set for the clustering, because of the redundancy in our full feature set and for visualization purposes. The tierpsy_256 set is a useful subset of the total feature set derived in a previous paper (Javer *et al*, 2018b), which we use as a starting point before doing problem-specific feature selection. For the hierarchical clustering, we create a matrix where each row is the average of all the replicates of a specific drug dose and each column is one of the tierpsy_256 features at a specific blue light condition. We also include 6 rows with the average of 6 partitions of the DMSO control replicates randomly samples across tracking days. We generate multiple DMSO averages to check how well they cluster together and locate the region of low compound effects in the clustermap. We create 6 DMSO partitions because in this way the number of averaged DMSO replicates is similar to the number of data points per compound. We use the hierarchical clustering algorithm implemented in the seaborn (Waskom, 2021) clustermap function with complete linkage and cosine distances. To assess the quality of the clustering, we use the row linkage dendrogram and compare the cluster purity at each level of the dendrogram with the purity of random clusters. To get the random clusters, we permute the cluster

labels derived from the dendrogram. We repeat the permutation 1,000 times. A flow chart of the steps followed for the hierarchical clustering is shown in Fig EV6.

### Classification by mode of action

The classification of compounds into modes of action is a classification of "bags" of data points, as we know that we have the same label across doses and across replicates of the same compound. To get compound-level predictions, we use the following approach. We train a single classifier with the rescaled bootstrapped averages from all the compounds, and then, we combine the predictions across all the data points of the same compound with a simple voting procedure. Each bootstrapped average point contributes one vote for the class of the compound. The predicted class for the compound is the one with the most votes. We initially tested three types of classifiers: logistic regression, random forest, and XGBoost. Logistic regression performed better than the ensemble methods in terms of cross-validation accuracy in the training set, so it was adopted for the entire pipeline. Despite being a linear classifier, logistic regression performs well in this classification problem probably because of the high dimensionality of the feature space, which renders the separation of classes with linear hyperplanes possible. At the same time, it limits overfitting both with the linear boundaries and with the adoption of regularization.

As we have thousands of highly correlated features, we performed feature selection with recursive feature elimination using the training set. The estimator used for feature selection was multinomial logistic regression with elastic net as implemented in sklearn (Pedregosa *et al*, 2011). The term elastic net refers to a combination of l1 and l2 regularization. We used a ratio of 0.5 between the l1 and l2 regularization parameters. We tested different sizes of selected feature sets and chose the set with 1,024 features, which resulted in the highest fourfold cross-validation accuracy. The full list of selected features is reported in Dataset EV2.

Using only the 1,024 selected features, we optimized the hyperparameters of the estimator using stratified fourfold cross-validation. We tested the parameter grid shown in Table 1. The optimal parameters for the logistic regression classifier are penalty="l2", C=10, multi_class="multinomial". The cross-validation accuracy presented in Fig 3B is the result obtained with the optimal parameters. Finally, using the optimal feature set and the optimal hyperparameters we trained a classifier with the entire training set and predicted the class of the unseen compounds in the held-out test set. The entire pipeline for the feature selection, hyperparameter tuning, training, and testing of the classifier is outlined in the flow chart in Fig EV7.

To obtain a baseline for our classification accuracy, we generated a null model by partitioning the DMSO data randomly across

**Table 1. Parameter grid for model selection**

| Parameter | Values option 1 | Values option 2 |
|---|---|---|
| Penalty | "l1", "l2" | "elasticnet" |
| C | 0.01, 0.1, 1, 10, 100, 1,000 | 0.01, 0.1, 1, 10, 100, 1,000 |
| L1_ratio | | 0.1, 0.5, 0.9 |
| Multi_class | "multinomial", "ovr" | "multinomial", "ovr" |
| Solver | 'saga' | "saga" |

tracking days to fictitious 10 classes. We then followed the same pipeline as for the real data: split the data in a training/tuning set and a test set, performed features and hyperparameter selection using cross-validation on the training set, trained a classifier with the selected features and hyperparameters on the entire training set, and predicted the class in the test set.

### Novelty detection

To detect potentially novel modes of action that are not part of the known classes seen by the trained classifier, we use an adjusted version of the novelty detection algorithm for multiclass problems proposed in Ref preprint: Vinokurov and Weinshall (2016). The algorithm uses θ scores to assess the affinity of a compound to the known classes. The θ score is defined as the ratio between the confidence of the classifier in the most likely class and the confidence in the second most likely class. Using our logistic regression classifier, the θ score of a compound is given by:

$$\theta = \frac{sort\left(\underset{i \in S}{mean}(class\_probas)\right)[0]}{sort\left(\underset{i \in S}{mean}(class\_probas)\right)[1]}$$

where $S$ is the set of data points belonging to the given compound and *class_probas* are the probabilities of each class predicted by the trained classifier. These θ scores are fed to an ensemble of binary support vector machines (SVMs) to flag novel compounds. The ensemble is trained with the following procedure. First, we partition the training set ten times, each time leaving one class out as presumed novel. For each partition, we train a logistic regression classifier and we get the θ scores for the compounds of the known classes and for the compounds of the presumed novel class. Using the θ scores and the mean θ scores in the respective class as input features, we train a binary SVM with RBF kernel to label compounds with 0 if they are known and with 1 if they are novel. This procedure gives us an ensemble of ten SVMs that take as input the θ score of a compound and the mean θ score of its assigned class and give as output a prediction of whether the compound is novel. The final novelty score of the compound is given by the average of the ten predictions weighted by the cross-validation accuracy of the SVM in its respective training set. A flow chart of the novelty detection method is shown in Fig EV8.

### Within mode-of-action classification

To test for phenotypic differences within mode-of-action classes, we trained a classifier to distinguish the replicates from each compound within a class from the replicates of the other compounds in the class. In this case, we did not use the bootstrapped averages, but the standardized and normalized raw data. For each of the ten modes of action included in the main classification task, we pool all the compounds from the training set and the test set. Each compound is considered a separate class. We use stratified fourfold cross-validation to include replicates of every compound at every dose both in the training and the test set in each fold. We then pool together the predictions for each test fold to create a confusion matrix for all the replicates. Finally, we cluster the confusion matrix using the spectral co-clustering algorithm as implemented in scikit-learn to reveal internal structure within the mode of action.

## Data availability

The datasets and computer code produced in this study are available in the following databases:

- Tierpsy features and metadata: Zenodo (https://doi.org/10.5281/zenodo.4681682).
- Raw data and tracking data: Zenodo (individual links per tracked multiwell plate reported in Dataset EV3).
- Analysis computer scripts in Python: GitHub (https://github.com/Tierpsy/moaclassification).

**Expanded View** for this article is available online.

### Acknowledgements

This project has received funding from the European Research Council (ERC) under the European Union's Horizon 2020 Research and Innovation Program (Grant Agreement No. 714853) and was supported by the Medical Research Council through Grant MC-A658-5TY30. AMR was supported by a BBSRC CASE studentship part-funded by Syngenta.

### Author contributions

AEXB, PHH, and AJF conceived the experiments. AMR, IB, and LF designed and conducted experiments. EM and AEXB conceived modeling. EM involved in the development of *moaclassification* and *tierpsytools* Python packages. LF and EM developed software to process tracking data. EM and LF produced figures. AEXB, EM, IB, AMR, and LF wrote the manuscript.

### Conflict of interest

Research grant support was provided by the Biotechnology and Biological Sciences Research Council of the UK in partnership with Syngenta UK. AJF and PHH are employees of Syngenta UK. AEXB has consulted with Syngenta UK.

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
