## [Review Process File · Molecular Systems Biology]

Behavioral fingerprints predict insecticide and anthelmintic mode of action

Adam McDermott-Rouse, Eleni Minga, Ida Barlow, Luigi Feriani, Philippa Harlow, Anthony Flemming, and André Brown

DOI: [10.15252/msb.202110267](https://doi.org/10.15252/msb.202110267)

Corresponding author(s): André Brown (andre.brown@imperial.ac.uk)

Review Timeline:

Submission Date:	2nd Feb 21
Editorial Decision:	11th Mar 21
Revision Received:	19th Apr 21
Editorial Decision:	20th Apr 21
Revision Received:	21st Apr 21
Accepted:	22nd Apr 21

Editor: Maria Polychronidou

Transaction Report:

Thank you again for submitting your work to Molecular Systems Biology. We have now heard back from the three referees who agreed to evaluate your study. Overall, the reviewers acknowledge that the presented approach is a relevant contribution to the field. They raise however a series of concerns, which we would ask you to address in a revision.

I think that the recommendations of the reviewers are rather clear. Therefore, I do not see the need to repeat the points listed below. All issues raised by the reviewers need to be satisfactorily addressed. Please contact me in case you would like to discuss in further detail any of the issues raised.

On a more editorial level, we would ask you to address the following points:

REFeree REPORTS

Reviewer #1:

The authors present their study entitled "Behavioral fingerprints predict insecticide and antihelmintic mode of action" by McDermott-Rouse et. al. In this report, the authors have engineered a novel imaging system consisting of an array of high resolution/high frame rate CMOS cameras to image an entire 96-well plate of nematodes incubated with various compounds and assessed the compound effect by multivariate measures. The authors conclude that this system could be used to identify novel insecticides in a high throughput manner. In general, the manuscript

is well written and the conclusions are solid. There are however some points that would aid the manuscript.

1. The authors utilize the model organism *C. elegans*. Since this model organism is routinely used to address biological questions, particularly using fluorescence, the authors could address whether their system would be adaptable to fluorescence imaging?

2. The authors utilize 96 well plates, however, most high-throughput screening platforms utilize 386-well or 1536-well plates. Could their system be adaptable to a more high-throughput system?

3. The authors state that 3 animals were placed per well. How was this number determined? was this experimentally derived? If yes, what would be the minimum and maximum number of *C. elegans* per well that could be used? If not, how was 3 animals per well derived?

4. In figure 2A, it would aid the reader if particular clusters could be highlighted, for example AChE inhibitors

5. In figure 2C, dose responses of several compounds are shown and the authors state "The three mitochondrial inhibitors in Fig. 2C all decrease angular velocity..." (lines 156-157). However, no statistics or experimental replicate information is provided. This figure requires statistical evaluation to support the conclusions in the text.

Minor points:

Symptomology (lines 20, 66, 288, 315) and the repeated use of symptoms is uncharacteristic of studies outside of patients. Usually, researchers will use "phenotype" to describe visual analyses of model organisms.

Reviewer #2:

Manuscript review report

Manuscript Title:

Behavioral fingerprints predict insecticide and anthelmintic mode of action

Summary

***Describe your understanding of the story

The claim is in the title: the authors describe the use of informatic methods to investigate mode of action of invertebrate-targeting compounds by high throughput screening/imaging by quantitative phenotyping of organism *C. elegans*. These methods use digital descriptors of behavioral phenotypes to cluster the effects of different compounds.

***What are the key conclusions: specific findings and concepts

The key claim is that the digitized behavioral fingerprints that are analyzed using the informatic methods described in the manuscript can be used to distinguish MOA classes via machine learning models and to also predict MOAs of previously unseen compounds.

However, I was not able to decipher the novel MOA or even what was the evidence for some very interesting sounding conclusions: e.g. towards the end of the introduction the authors state "Finally, we show that our prediction accuracy may not be limited by noise or phenotypic dimensionality." This leads the authors to conclude the digital clustering of behaviors can uncover a more refined mode of action than currently understood by the pharmacology.

***What were the methodology and model system used in this study

They previously developed pioneering imaging and image analysis methods to obtain high throughput imaging of *C. elegans* behaviors. In this study, they expand on these methods to develop behavioral fingerprints following the addition of compounds that are known to act as insecticides and anthelmintic compounds through a diverse array of modes of action. They use a combination with a number of compounds to train and test a machine learning model, the model is trained with a minimum of 4 compounds per class, and 10 compounds for the classification task. 60 compounds were used in the training set, and 16 compounds were used as a test set.

*General remarks

***Are you convinced of the key conclusions?

The claims of the manuscript are innovative and significant. Unfortunately, the manuscript was difficult to follow and with a number of lacunae to be able to fully assess the validity of the conclusions, these could be remedies but hard to assess some of the conclusions as discussed here. I draw your attention to the commentary published in this journal (<https://doi.org/10.15252/msb.20209982>) - as written, this manuscript does not pass the tests mentioned there.

***Place the work in its context.

Digital readouts of complex phenotypes - in this case, behavioral and postural phenotypes of *C. elegans*, are of critical importance and the authors have pioneered some of that development. This work explores the use of these data in understanding and predicting mode of action of biologically active compounds.

***What is the nature of the advance (conceptual, technical, clinical)?

This is a mostly technical advance, with some possibility of discovering new modes but that part is less clear. Mode of action is equated to a form of pheno-clustering and this could be described better. Interested readers might think of mode of action descriptions to have to reach a deeper mechanistic level than this study provides.

***How significant is the advance compared to previous knowledge?

The type of work is both critical and interesting - previous knowledge in areas of classifying phenotypes is the bedrock of genetic screening but this more digital formalism has been developing for over a decade. This study does show that behavioral phenotypes can provide enough information to cluster molecules. It is an important advance.

***What audience will be interested in this study?

Those doing primary screens for any perturbation - genetic, chemical, environmental etc.

*Major points

***Specific criticisms related to key conclusions

The authors use a set of compounds with 'sparsely populated classes' and they label them as "novel compounds", these are used for novelty detection. The 17 novel compounds span include 11 classes.

I am trying to find (in the manuscript) where an unknown compound's MOA was discovered? This would be a major finding, but the manuscript doesn't elaborate on the "novel" compounds, and there is no mention of specific features/phenotypes and how they connect to the mode of action of these novel/unknown compounds.

Should these compounds be called "novel", are they unknown compounds? Perhaps they should be called "test" compounds, if they aren't actually novel/unknown compounds?

Abstract line 24 - "we also classify compounds within each mode to discover pharmacological relationships"

What does this mean?

What pharmacological relationships were discovered?

Does this mean they classified compounds with unknown MOAs within each of the 10 MOAs? If so, which ones?

Were they further verified, how many unknowns were classified as having one of the 10 MOAs?

***Specify experiments or analyses required to demonstrate the conclusions

When training a machine learning model (classification task) the number of samples used will greatly affect the number of models that can fit the data, which often results in over-fitting. The authors don't discuss overfitting and how this is addressed, but they use cross validation and state their (regularization?) parameters "L2" and C = 10 as optimal for predictive accuracy.

Can they elaborate on this, as the number of predictors (1000+) far exceeds the number of samples/observations?

The authors don't use a large set of compounds, so it would be useful to repeat the model fitting (training, split, test) on DMSO behavioral fingerprints as a NULL model. Each DMSO well would be treated as a compound having a specific class.

This might provide some clues on model reliability. If the data is available, doing this would offer a comparison in model performance, or prediction accuracy. It would be a NULL model, with little structure, so should not perform well (I assume).

***Motivate your critique with relevant citations and argumentation

The literature, via simulations show that K-fold Cross-Validation (CV) produces strongly biased performance estimates with small sample sizes, and the bias is still evident with sample size of 1000

<https://journals.plos.org/plosone/article?id=10.1371/journal.pone.0224365>

The literature suggests, after assessing 568 fitted predictive models, between 80 to 560 annotated samples are needed.

<https://bmcmmedinformdecismak.biomedcentral.com/articles/10.1186/1472-6947-12-8>

*Minor points

***Easily addressable points

Behavioral fingerprints capture posture, motion and path

line 76 - how many dimensions were used?

o How many do they measure?

o How many do they end up using in each task and in each figure?

o Figure one uses ~ 250? Why?

o The image acquisition could use more detail - perhaps a drawing of beyond the "custom-built tracking rigs".

***Presentation and style

A flow chart or outline of data pre-processing and normalization steps would help follow the sequence of the analysis. (Z score normalization, Fingerprint averages by resampling, per Row scaling, train/test different machine learning models, multi-logistic regression model, feature selection, novelty detection algorithm, SVM classifiers, novel compounds)

Figure 1: Three compounds are mentioned - which have strong effects on *C. elegans*, they separate from DMSO and from each other in a 2D plot of speed/body curvature(1/r).

Question: fig 1B - why aren't they color coding the actual grey-data points by treatment condition, dose, replicate? Do the grey points include other compounds than the three mentioned? Do the points include replicates? What are the lines and crosses representing?

Question: fig 1D

Are they identifying # of significant features that are different from the control for each behavioral fingerprint? Is this step used for identifying compounds with no effect or is this a form of feature selection or data normalization?

Was this done comparing all DMSO vs. DMSO fingerprints as well? What would the DMSO heatmaps look like? Very dark colors (~small number of features)?

How many features are in each behavioral fingerprint? The max looks like 10^3 , are there over 1000 features here? Are the rows of each small heatmap different compounds or replicates and doses of one compound? Regarding the three additional stimuli - are they experimental conditions? Is this information/result used elsewhere in the paper?

Figure 2:

Question: fig 2A

It looks like DMSO (black) doesn't cluster together, why is that?

5 rows are considered "DMSO" but they are an average of 5 random partitions of the DMSO control replicates? Why do they do this? Why are there only 5 averaged-DMSO?

Is the bottom portion of the heatmap low dose or low activity?

Do the annotated-colored compound classes (rows) include several different compounds or replicates or both?

Is the range on Z score (-1, 1) or is the range wider?

Why do they use this specific set of 256 features?

Question: fig 2C

How many data points make up each box-plot? Are the distributions representative of one or

multiple wells? Are the dose response boxplots meant to support the clustering? Where in the cluster are these specific treatments?

How many data points in a well - how many worms per well/treatment condition?

Line 171 - Resampling - data smoothing

If there are several replicates and multiple doses, what is the motivation for smoothing/bootstrapping the data by "resampling with replacement"? Why not just average over the 10 replicates? Can the replicate data be merged into one larger sample? Is the average fingerprint meant to represent all doses and replicates of a particular compound? This data smoothing/manipulation must change the data quite a bit, can they authors show an example of before - and - after fingerprint?

Line 178 - Row normalization

This is the third data smoothing/scaling method so far, the methods could be ok, but it is hard to tell if they don't show a pre and post processing of the data, and over-manipulating the data could result in loss of information and misinterpretation of results.

Question: fig 3A

There appears to be more structure in the unscaled data, why don't the authors use their real data to show this scaling procedure it is not obvious if there are any benefits in doing this row-wise scaling.

Line 185 - I am not familiar with the multi sensor fusion problem

They train and predict replicates? Then why did they create average fingerprints?

They train on 60 compounds and test 16 compounds?

Training data was used to determine an appropriate classifier - do they mean, "machine learning model"? How many machine learning models did they test - just the three mentioned models? They chose the multinomial logistic regression model based on cross-validation accuracy, with 1024 features?

There is no discussion on the multinomial regression model after that, why does it work on your data?

Line 413: 3020 features + derived features and stimuli results in 9060 features per well.

Question: Which features are used where? Are the three stimuli used for the classification task?

Line 443 we normalize each behavioral fingerprint to unit L2-norm. Rescaling in this way brings compounds with similar effect profiles across features but different potencies closer together in feature space.

Question: Can they give an example where compounds have similar effect profiles but different potencies?

Reviewer #3:

Summary

The authors developed a pesticide drug classifier pipeline based on the effect drugs had on behavioral metrics in *C. elegans*. A variety of drugs were used with known targets to determine whether drugs with the same mode of action produced the same behavioral effects. Several drugs did indeed produce similar behavioral changes based on shared targets. The authors then leveraged this feature of their data to develop classifiers to predict a drug's mode of action based on its phenotypic effects. They furthered this analysis by comparing drugs with known targets, but differing sub-targets or off-targets, and showed certain drugs showed significantly different behavioral effects than other drugs that shared the same primary target, but different off-targets. The key findings of this paper are that *C. elegans* can be used as a valuable symptomatic model for testing the mode-of-action of a variety of drugs. The authors have sufficient behavioral resolution to distinguish different categories of drugs, even though many of them cause motor defects in *C. elegans*. However, their detailed list of behavioral features enables them to distinguish the different classes of drugs.

Their work is greatly enabled by the high-throughput pipeline they have developed to test the drugs. The liquid handling and small size of *C. elegans* allowed the researchers to test multiple drugs at different dosages and environmental conditions in a timely fashion.

General remarks

On the whole, I really enjoyed this paper. The authors have nicely extended the high-throughput behavioral analysis pipeline they have published in earlier work. As they mentioned, high-throughput symptomatic screening is challenging, especially if manual annotation is required. Their work has improved upon the standard pipeline by using machine vision, and a behavioral model that has limited complexity, but also a sufficient level of complexity to discriminate drug classes. People interested in drug development will be interested in this work, but also behavioral researchers using similar machine vision protocols to see potential applications that can be realized with these approaches.

Most of my comments are pretty minor:

1. What are the lines in Figure 1B? Confidence intervals, SD, quartiles?
2. There appear to be far more drugs in Figure 2A than the 76 used for the classifiers. How were these drugs chosen? It appears there are 10 drug classes in Figure 2A, and 10 drug classes in Figure 3. Was there a bias for drugs that classified well in Figure 2A? If I missed how the 76 were chosen, I apologize. The method section describes how the training and test sets were split, but I don't understand how the original 76 compounds were chosen in the first place.
3. I would have appreciated a more detailed explanation of the estimator they used, and what the final parameters were for their classifiers.
4. Is there raw data anywhere so readers can verify the analyses for themselves? This would be useful not only to verify the models, but also as a resource for the community.
5. I think there should be a clarification that the dosage concentrations mentioned in the paper are not necessarily the dosage experienced by the animals themselves, but the concentration of drugs in the plates. The cuticle of the animals forms a notoriously difficult barrier for many drugs. The availability of drugs to the animal may also be a source of variance in the data (which would contribute to differing dosage EC50s).

6. The following experiment isn't needed for the paper, but there are cuticle-mutant strains that have been used with better drug penetrance. For example, this paper:

<https://doi.org/10.1038/s41598-017-10454-3>

The usage of these strains may increase the number of drugs that have a behavioral effect on the animals, and/or decrease effective dosages.

I recommend this paper for publication.

Reviewer comments are in blue

Author responses are in black

Reviewer #1:

The authors present their study entitled "Behavioral fingerprints predict insecticide and antihelminthic mode of action" by McDermott-Rouse et. al. In this report, the authors have engineered a novel imaging system consisting of an array of high resolution/high frame rate CMOS cameras to image an entire 96-well plate of nematodes incubated with various compounds and assessed the compound effect by multivariate measures. The authors conclude that this system could be used to identify novel insecticides in a high throughput manner. In general, the manuscript is well written and the conclusions are solid. There are however some points that would aid the manuscript.

1. The authors utilize the model organism *C. elegans*. Since this model organism is routinely used to address biological questions, particularly using fluorescence, the authors could address whether their system would be adaptable to fluorescence imaging?

We think it can be and we are currently working on implementing fluorescence imaging on these rigs but the work is at an early stage.

2. The authors utilize 96 well plates, however, most high-throughput screening platforms utilize 386-well or 1536-well plates. Could their system be adaptable to a more high-throughput system?

1536-well plates are too small to support adult worm behaviour, but we do believe it will be possible to operate with 384-well plates. However, because of issues with parallax and because of the steep angle of the meniscus in 384-well plates this will require custom plates, which we are starting to work on.

3. The authors state that 3 animals were placed per well. How was this number determined? was this experimentally derived? If yes, what would be the minimum and maximum number of *C. elegans* per well that could be used? If not, how was 3 animals per well derived?

In the first round of imaging (December 2019) we used 2 worms per well, and moved to 3 in the second (January 2020). This was missing from the original manuscript, and we are rectifying it now. Our goal was to minimise worm-worm contacts which complicate tracking while ensuring that there is usually at least one worm visible for tracking. Increasing the worm density further increases the probability of worm-worm contact, leading to shorter tracks (identity is lost upon contact) and more frames without pose estimation (skeletonising worms in contact is problematic).

Swierczek *et al.* (<https://doi.org/10.1038/nmeth.1625>) showed (fig 1e) that increasing the worm density has diminishing returns in terms of fraction of poses quantified. With 2 worms per well (useful imaging area 40mm²) the worm density in our system is approximately the same as having 100 worms in a 5 cm Petri plate in the Swierczek *et al.* setup. We have

4. In figure 2A, It would aid the reader if particular clusters could be highlighted, for example AChE inhibitors

We are concerned that highlighting classes might be confusing. We want the reader to notice the obvious clusters, but also notice that even in those classes that cluster well some doses of certain compounds are far away from the obvious cluster. We think that the colour-coding of rows according to class (at the left side of the heatmap) helps to explain the distinct blocks of the clustermap. Additionally, we have added a second bar of row colours at the right side of the heatmap to mark the locations of the compounds plotted in Fig 2C. By highlighting the locations of these compounds, we give two representative examples of how closely different doses of the same compound cluster and how closely different drugs of the same class cluster. The three spiroindolines (vAChT inhibitors) constitute an example of a class that clusters well, since all of the compounds of the class are potent. On the contrary, the mitochondrial inhibitors contain a mix of more potent and less potent compounds, which complicates simple clustering of this class.

5. In figure 2C, dose responses of several compounds are shown and the authors state "The three mitochondrial inhibitors in Fig. 2C all decrease angular velocity..." (lines 156-157). However, no statistics or experimental replicate information is provided. This figure requires statistical evaluation to support the conclusions in the text.

We have included the individual replicates (i.e the number of wells per condition) in the figure. We also included details of the statistic tests used in the legend of each plot. These tests are based on linear mixed models, with the drug dose as fixed effect and the experiment day as random effect. The results cited are the p-values and the coefficients for the fixed effect.

Minor points:

Symptomology (lines 20, 66, 288, 315) and the repeated use of symptoms is uncharacteristic of studies outside of patients. Usually, researchers will use "phenotype" to describe visual analyses of model organisms.

This terminology also came as a surprise to some of us, but 'symptoms' is widely used in the insecticide community (see for example reference 6). We would prefer to continue using symptoms in the introduction to appeal to this community.

Reviewer #2:

Manuscript review report

Manuscript Title:

Behavioral fingerprints predict insecticide and anthelmintic mode of action

Summary

***Describe your understanding of the story

The claim is in the title: the authors describe the use of informatic methods to investigate mode of action of invertebrate-targeting compounds by high throughput screening/imaging by quantitative phenotyping of organism *C. elegans*. These methods use digital descriptors of behavioral phenotypes to cluster the effects of different compounds.

***What are the key conclusions: specific findings and concepts

The key claim is that the digitized behavioral fingerprints that are analyzed using the informatic methods described in the manuscript can be used to distinguish MOA classes via machine learning models and to also predict MOAs of previously unseen compounds. However, I was not able to decipher the novel MOA or even what was the evidence for some very interesting sounding conclusions: e.g. towards the end of the introduction the authors state "Finally, we show that our prediction accuracy may not be limited by noise or phenotypic dimensionality." This leads the authors to conclude the digital clustering of behaviors can uncover a more refined mode of action than currently understood by the pharmacology.

We have clarified these points in the text. We provide more details in our response to the specific comments below.

***What were the methodology and model system used in this study

They previously developed pioneering imaging and image analysis methods to obtain high throughput imaging of *C. elegans* behaviors. In this study, they expand on these methods to develop behavioral fingerprints following the addition of compounds that are known to act as insecticides and anthelmintic compounds through a diverse array of modes of action. They use a combination with a number of compounds to train and test a machine learning model, the model is trained with a minimum of 4 compounds per class, and 10 compounds for the classification task. 60 compounds were used in the training set, and 16 compounds were used as a test set.

*General remarks

***Are you convinced of the key conclusions?

The claims of the manuscript are innovative and significant. Unfortunately, the manuscript was difficult to follow and with a number of lacunae to be able to fully assess the validity of the conclusions, these could be remedies but hard to assess some of the conclusions as discussed here. I draw your attention to the commentary published in this journal (<https://doi.org/10.15252/msb.20209982>) - as written, this manuscript does not pass the tests mentioned there.

We have included more details on the methodology in the methods and text and specifically address the points raised in the comments below. We agree on the issue of reproducibility and emphasise that all of the associate code with extensive comments and documentation is available online. This includes not just the basic code underlying the analysis but also scripts specifically to reproduce all of the figures (<https://github.com/Tierpsy/moaclassification>).

The commentary addresses the issue of the reproducibility of simulations based on ODE models so some of the points do not apply here (for example the specification of initial conditions and the use of standardised modelling frameworks). We made sure that the points that apply to our case are satisfied. The methods that we propose are based on standard

implementations of machine learning algorithms available in scikit-learn. The only mathematical expression we implemented independently is the expression for the novelty score, which is described in the methods. The implementation of the expression is included in the published code, properly commented. The most analogous quantities to the model parameters and initial conditions of ODE models in our case are the hyper-parameters of the clustering algorithm and the classifiers. We included all the significant hyper-parameter values in the methods. The complete set of hyper-parameters can also be found in the code that is publicly available on Github. In addition to the code, we have included our trained models. Finally, we have included a more detailed description of every step of our pipeline in the methods and flowcharts in the extended view to facilitate reproducibility in other programming languages.

***Place the work in its context.

Digital readouts of complex phenotypes - in this case, behavioral and postural phenotypes of *C. elegans*, are of critical importance and the authors have pioneered some of that development. This work explores the use of these data in understanding and predicting mode of action of biologically active compounds.

***What is the nature of the advance (conceptual, technical, clinical)?

This is a mostly technical advance, with some possibility of discovering new modes but that part is less clear. Mode of action is equated to a form of pheno-clustering and this could be described better. Interested readers might think of mode of action descriptions to have to reach a deeper mechanistic level than this study provides.

***How significant is the advance compared to previous knowledge?

The type of work is both critical and interesting - previous knowledge in areas of classifying phenotypes is the bedrock of genetic screening but this more digital formalism has been developing for over a decade. This study does show that behavioral phenotypes can provide enough information to cluster molecules. It is an important advance.

***What audience will be interested in this study?

Those doing primary screens for any perturbation - genetic, chemical, environmental etc.

*Major points

***Specific criticisms related to key conclusions

The authors use a set of compounds with 'sparsely populated classes' and they label them as "novel compounds", these are used for novelty detection. The 17 novel compounds span include 11 classes.

I am trying to find (in the manuscript) where an unknown compound's MOA was discovered? This would be a major finding, but the manuscript doesn't elaborate on the "novel" compounds, and there is no mention of specific features/phenotypes and how they connect to the mode of action of these novel/unknown compounds.

Should these compounds be called "novel", are they unknown compounds? Perhaps they should be called "test" compounds, if they aren't actually novel/unknown compounds?

We have updated the text to make the distinction between training/tuning data, test data, and "novel" test data. In our library we used the well-represented mode of action classes (those

with more than four compounds) to train a classifier. These are the 10 classes seen in Figures 3(b) and 3(c). We had 76 compounds spanning these 10 classes and we split these in a training/tuning set and a test set to perform the classification task (i.e. check how accurately we can predict the mode of action of a compound that belongs to one of these 10 classes).

The remaining compounds in our library (17 compounds spanning 11 mode of action classes) were assigned to what we have now called “novel test set”. These compounds have known modes of action, but these are distinct from the 10 mode of action classes that our classifier has been trained on. In other words, the mode of action classes of the novel test set have not been seen by our classifier. Therefore, our classifier by construction cannot predict the mode of action of these compounds.

However, we wanted to see if we can use the predictions of the classifier to get an indication of whether a given compound belongs to the set of classes the classifier has been trained on. This is what our novelty detection method is designed to do: detect whether a compound belongs to one of the 10 modes of action or not. This can be seen as a measure of confidence in the classifier prediction.

In practice, we hope that this method could facilitate drug discovery by prioritising compounds as lead candidates for further investigation. We imagine a situation where we don't know the mode of action of a new compound and we want to know whether it belongs to one of our major known classes or not. If it doesn't belong to one of the known classes, we want to investigate it further as a possible lead. If we have a classifier trained to predict our major known classes, we can use our novelty detection method to try and answer this question. A high novelty score would encourage us to prioritize this compound for further investigation.

Of course, we hope that this test will translate to a situation where we have a truly novel mode of action that has never been previously described. However, the discovery of compounds with new modes of action is rare even in industrial screening facilities and so this will only be possible to test in a time scale of years.

Abstract line 24 - "we also classify compounds within each mode to discover pharmacological relationships"

What does this mean?

What pharmacological relationships were discovered?

Does this mean they classified compounds with unknown MOAs within each of the 10 MOAs? If so, which ones?

Were they further verified, how many unknowns were classified as having one of the 10 MOAs?

We trained our classifier to predict known classes using the training set, which consists of 60 compounds with 10 different modes of action. We tested the accuracy of the trained classifier in the test set which consists of 16 compounds spanning the same 10 modes of action. This classification task was based on the assumption that the compounds that have the same primary target (i.e. belong to the same mode of action class) will have similar behavioural fingerprints. Our prediction accuracy confirmed this hypothesis.

However, we were aware that the effects of the compounds used in this work were more complex; in addition to the primary mode of action, they can have more specific modes of

action or known off-target effects. Therefore, we wanted to investigate whether there is substructure within the mode of action classes. To do that, we performed a different classification task: using data from a single mode of action each time, we tried to classify individual compounds. In this case, the null hypothesis would be that compounds with the same mode of action would have indistinguishable effects. Instead, we find that we can distinguish individual compounds within a class. We don't know whether these differences are due to pharmacokinetic or pharmacodynamic differences (or both).

The second result from the within-mode-of-action classification is that in some classes there are subsets of compounds that are more difficult to distinguish among each other. In a couple of cases illustrated in Fig. 4, these subsets correspond to previously known subgroups within a primary mode of action. So we can detect subgroups that reflect previously known 'pharmacological relationships'. We hypothesise that the other subgroups may also reflect currently unknown substructures within the primary modes of action.

To avoid misinterpretation, we have removed the reference to pharmacological relationships and made the abstract more precise.

***Specify experiments or analyses required to demonstrate the conclusions

When training a machine learning model (classification task) the number of samples used will greatly affect the number of models that can fit the data, which often results in overfitting. The authors don't discuss overfitting and how this is addressed, but they use cross validation and state their (regularization?) parameters "L2" and $C = 10$ as optimal for predictive accuracy.

Can they elaborate on this, as the number of predictors (1000+) far exceeds the number of samples/observations?

We agree with the reviewer that overfitting can be a serious problem in this classification task, because of the large number of predictors compared to the number of samples. For this reason, we took several steps to guard against overfitting.

Firstly, we select a subset of our full feature set (which includes more than 9000 predictors) using recursive feature elimination. This method discards redundant or less relevant features that might lead to overfitting of our trained model. Secondly, we use L2 regularization in our model, which further controls overfitting by penalising excessive model complexity. Both the feature selection and the tuning of the model hyperparameters (including the penalty parameter for the regularization) is done using cross-validation, as the reviewer states. This is an established preventive measure for overfitting, as the model tuning is done based on the prediction accuracy in the held-out folds. Importantly, because we fit several models and tuned multiple hyperparameters when choosing the model, there is a risk that the cross-validation results are over-optimistic. For that reason, we used a held-out test set of compounds that were not seen at all during the training/tuning phase to check how well the trained model generalises to data outside the training set. The accuracy on predicting this unseen test set is the one we report in the abstract. The test set accuracy is very similar to the cross-validation accuracy, which confirms that we do not have a problem of overfitting. We have added a discussion about overfitting and the measures we took to prevent it in the methods. We also report all the model parameters tested and the ones finally selected.

The authors don't use a large set of compounds, so it would be useful to repeat the model fitting (training, split, test) on DMSO behavioral fingerprints as a NULL model. Each DMSO well would be treated as a compound having a specific class.

This might provide some clues on model reliability. If the data is available, doing this would offer a comparison in model performance, or prediction accuracy. It would be a NULL model, with little structure, so should not perform well (I assume).

We have performed the DMSO classification as suggested. We assigned DMSO points to 10 fictitious classes. Each class contained 48 DMSO replicates, randomly chosen across all plates and tracking days. We split the data in a training set and test set and we followed exactly the same pipeline as in the actual drug classification task. First, we selected features and tuned model parameters using cross-validation in the training set. The highest cross-validation accuracy achieved was 10% (equivalent to random choice). Then we trained a model with the selected features and hyperparameters and made class predictions in the test set. The test accuracy was 12.4%, which can be considered our baseline accuracy based on this null model. We added this result to the text.

We also devised a second null model, by randomly shuffling the class labels of the 76 compounds that are included in the MOA classification task. We then followed the same pipeline as in the real classification task and obtained a maximum cross-validation accuracy of 14% and a test accuracy of 12.5%.

These two null models provide baselines for the classification and help illustrate that there is a strong real signal in our data. The scripts to get the results for the null models have been added to the github repo.

***Motivate your critique with relevant citations and argumentation

The literature, via simulations show that K-fold Cross-Validation (CV) produces strongly biased performance estimates with small sample sizes, and the bias is still evident with sample size of 1000

<https://journals.plos.org/plosone/article?id=10.1371/journal.pone.0224365>

We appreciate the factors reported in this paper and indeed we combine the two approaches recommended in the paper for determining our reported accuracy: 1) the CV accuracy we report in Figure 3B is obtained using nested CV (we select the best number of features and we tune the hyperparameters of the model using a nested approach, although we do not use the term *nested* as we consider this approach standard practice for model selection using CV) and 2) we use a train/test split and report the test accuracy in Figure 3C. As mentioned in the methods, the reason we chose only the 10 classes with more than 4 compounds was to be able to make this train-test split, so we can report both cross-validation and test accuracy. Based on the results of the paper referenced by the reviewer, both our accuracy estimates should be unbiased. Indeed, the fact that our CV and test accuracy are similar (Figure 3C), indicates that our estimate generalises well in unseen data and is therefore reliable.

The literature suggests, after assessing 568 fitted predictive models, between 80 to 560 annotated samples are needed.

<https://bmcmmedinformdecismak.biomedcentral.com/articles/10.1186/1472-6947-12-8>

This study proposes a method for predicting the stopping point for annotations in active learning models and the results in terms of number of annotated samples are not meant as general guidelines for classification problems. The performance of a classifier with a specific sample size depends on myriad parameters, including but not limited to the type of classifier, the type and number of features, the features to samples ratio, the model hyperparameters and the number of classes. We employed standard methods to get accuracy estimates that are unbiased, as discussed in the previous comment. Therefore, we are confident that we have a real signal adequately represented by our sample. It should also be noted that we train our classifier using all the bootstrapped averaged data points, which means that the number of annotated data points used to train the classifier is in the range of 9000.

*Minor points

***Easily addressable points

Behavioral fingerprints capture posture, motion and path

line 76 - how many dimensions were used?

o How many do they measure?

o How many do they end up using in each task and in each figure?

o Figure one uses ~ 250? Why?

We have clarified these points in the text and the methods. We measure 3020 features in total from each video. These features are described in detail in the paper that presents tierpsy tracker [ref 23]. We then treat the features from the pre-stimulus, blue light, and post-stimulus videos as independent features, which gives us a total of 9060 features per well. Many of these features are correlated with each other. The actual number of independent dimensions is much smaller. PCA suggests that approximately 2000 dimensions explain 99% of the total variance.

The data in the first clustering figure is analysed before any feature selection on the data collected here. In this case, we used 256 features per video (a total of 768 features). These 256 features (referred to as the tierpsy_256 below) were selected in a previous paper [ref 24] and we use them routinely as a more manageable starting point for new analyses.

For the classification task in this paper we used the training data to select a feature set specific for the drug classification task. We selected the optimal number of features among four possible options $\{2^n, \text{ for } n \text{ in } 7:10\}$ using cross validation (the possible sizes are based on the results from the PCA shown above). The optimal number turned out to be 1024 features. We then selected a set of 1024 features using the entire training for the subsequent

analyses. We have added the specific feature numbers to the captions in each figure where applicable.

o The image acquisition could use more detail - perhaps a drawing of beyond the "custom-built tracking rigs".

We have addressed the lack of details about the tracking rigs by adding detailed information about the imaging hardware and software in the methods section of the manuscript.

***Presentation and style

A flow chart or outline of data pre-processing and normalization steps would help follow the sequence of the analysis. (Z score normalization, Fingerprint averages by resampling, per Row scaling, train/test different machine learning models, multi-logistic regression model, feature selection, novelty detection algorithm, SVM classifiers, novel compounds)

We have added flow charts as Extended View figures describing: the pre-processing of the data, the hierarchical clustering in Figure 2, the classification and the novelty detection method.

Figure 1: Three compounds are mentioned - which have strong effects on *C. elegans*, they separate from DMSO and from each other in a 2D plot of speed/body curvature(1/r).

Question: fig 1B - why aren't they color coding the actual grey-data points by treatment condition, dose, replicate? Do the grey points include other compounds than the three mentioned? Do the points include replicates? What are the lines and crosses representing?

Each of the grey points represents the mean of the replicates of a compound at a particular dose. The points highlighted in colour are the selected compounds at a single dose, as indicated in the legend of the plot. We highlight only these points as an example of three compounds that can be separated easily at specific doses. We include all the remaining compounds at all their doses to show that in most cases this separation is not easy in a 2D speed-curvature space.

The crosses show the standard deviation in each dimension. We have added this information in the figure legend. We have also added the standard deviation bars to all the grey points to make it clear that they have the same status as the highlighted points.

Question: fig 1D

Are they identifying # of significant features that are different from the control for each behavioral fingerprint? Is this step used for identifying compounds with no effect or is this a form of feature selection or data normalization?

The heatmap shows the number of significant features for each compound compared to control. It is used to identify compounds with no effect not for normalisation. This step corresponds to the "Filter compounds based on univariate statistical tests" process outlined in the pre-processing flowchart that we added to the extended view.

Was this done comparing all DMSO vs. DMSO fingerprints as well? What would the DMSO heatmaps look like? Very dark colors (~small number of features)?

The test for each compound includes all of the replicates for that compound compared to all of the replicates for DMSO. If we subdivide the DMSO replicates as done above for classification and treat them similarly to the compounds we indeed see no significant differences, which would correspond to the darkest color in the heatmap colorbar in Figure 1 for all three conditions.

How many features are in each behavioral fingerprint? The max looks like 10^3 , are there over 1000 features here? Are the rows of each small heatmap different compounds or replicates and doses of one compound?

We used all 9060 features for each comparison. Each row is a different compound. All of the replicates are used in the statistical test and the linear model simultaneously considers all of the doses. For each univariate test, a significant effect is reported when the p-value for the drug dose is smaller than 0.01 after correcting for multiple comparisons using the Benjamini-Yekutieli method.

Regarding the three additional stimuli - are they experimental conditions? Is this information/result used elsewhere in the paper?

Each column represents the results from the three videos that were recorded for each sample, a pre-stimulus video, a blue light video, and a post-stimulus video. The heatmap shows that the blue light video is best at distinguishing compound treated animals from control (the middle column tends to be lighter than the other two). Since different features are different in each of the conditions we chose to treat the features from each video independently, concatenating the feature vectors from each video into a larger 9060 feature vector. This larger vector is the one used in all downstream analyses. For the clustering in Figure 2, we concatenate the tierpsy_256 features from each condition. For the classification task, we perform feature selection starting from the full 9060-dimensional vector. Details about the three recordings (pre-stimulus, blue light and post-stimulus) and how we use the features from all the videos can be found in the Methods in the last paragraph of section "Image acquisition" and in the first paragraph of section "Feature extraction and pre-processing". We also mention these experimental conditions in the Results section in the last paragraph of the section "*Insecticides affect phenotypes in multiple behavioral dimensions*", where we explain that the motivation behind using blue light.

Figure 2:

Question: fig 2A

It looks like DMSO (black) doesn't cluster together, why is that?

5 rows are considered "DMSO" but they are an average of 5 random partitions of the DMSO control replicates? Why do they do this? Why are there only 5 averaged-DMSO?

Is the bottom portion of the heatmap low dose or low activity?

In the initial heatmap, we included 5 subsets of DMSO points, each representing a different tracking day. Upon reflexion based on the comment of the reviewer, we have changed this to 6 subsets of DMSO points randomly sampled (without replacement) across all tracking days. We chose to create multiple subsets of DMSO to be able to see if they cluster together. We chose 6 subsets because in this way the number of averaged DMSO replicates is similar to the number of data points per compound.

We consider that this new partitioning is a better representation of the DMSO averages than the one we were using before, because the compound replicates are randomized across different days. Therefore, the DMSO averages should also be randomized and not reflect the day-to-day variation.

In the original version, the bottom portion of the heatmap was the set of low dose or low activity groups and the DMSO points were basically distributed randomly through this set, reflecting the day-to-day variation in the experiments. Using the new partitioning of DMSO points, we observe close clustering among them at the centre of the portion of the heatmap that represents low dose and low activity (which is now the top part of the heatmap). We have clarified this in the text and the methods.

Do the annotated-colored compound classes (rows) include several different compounds or replicates or both?

Each row is the average of all of the replicates of a given dose, as stated in the methods and as now emphasised in the caption and the text.

Is the range on Z score (-1, 1) or is the range wider?

The display range is from -1 to 1 to help visualise the clusters. The full range of z-scores is wider.

Why do they use this specific set of 256 features?

These features were reported in a previous paper [ref 24] as a useful subset of the total feature set which we use as a starting point for new analysis before doing problem-specific feature selection. We have clarified this in more detail in the Methods.

Question: fig 2C

How many data points make up each box-plot? Are the distributions representative of one or multiple wells? Are the dose response boxplots meant to support the clustering? Where in the cluster are these specific treatments?

We have added points to the box plots where each point represents a single well. With the box plots we want to illustrate some issues with dose response curves that can complicate clustering and that make it necessary to consider multiple doses. These issues are related to the different potencies of different compounds and to the non-monotonic effects observed in some compounds.

We have added a second bar with row colours at the right side of the heatmap to indicate the positions of the compounds/doses that are shown in Fig 2C. We matched the row colours to the colours of the boxplots.

How many data points in a well - how many worms per well/treatment condition?

Each well contains 2 or 3 worms. We treat each well as a single data point reporting the average of the worms in the well. We have included the number of wells per condition in the boxplot figures. On average, in our cleaned dataset we have 12 replicate wells per drug treatment and 601 DMSO control points.

Line 171 - Resampling - data smoothing

If there are several replicates and multiple doses, what is the motivation for smoothing/bootstrapping the data by "resampling with replacement"? Why not just average over the 10 replicates? Can the replicate data be merged into one larger sample? Is the average fingerprint meant to represent all doses and replicates of a particular compound? This data smoothing/manipulation must change the data quite a bit, can they authors show an example of before - and - after fingerprint?

The smoothing is done to reduce the impact of outlying wells on the classifier during training. Simply taking the average of the samples also reduces the effect of outliers but then the classifier loses information on the distribution of points. Additionally, taking bootstrapped averages allows us to balance the number of data points in each class, which reduces the risk of the classifier overfitting to the large classes. We tested all three approaches (no smoothing, averaging, and smoothing) and found the best results using the smoothing approach we reported. We have made these steps more explicit in the main text and methods and added the following figures that show the effect of smoothing and balancing in the most and least populated classes in the extended view. As can be seen in these figures, the fingerprints do not change in a significant way. Rather, the effect of outliers is reduced and the region of the phenotypic space corresponding to a specific class is populated.

Line 178 - Row normalization

This is the third data smoothing/scaling method so far, the methods could be ok, but it is hard to tell if they don't show a pre and post processing of the data, and over-manipulating the data could result in loss of information and misinterpretation of results.

All of these steps were tried on the training data set and not the test data so while there is a risk that the cross-validation performance could be inflated, the test accuracy should still be reliable.

Question: fig 3A

There appears to be more structure in the unscaled data, why don't the authors use their real data to show this scaling procedure it is not obvious if there are any benefits in doing this row-wise scaling.

We opted for the cartoon data for illustration for two reasons. Firstly, the real data do not separate well in three dimensions and so the plots would only be interpretable with a small hand-selected set. Secondly, the cartoon plot gives us the opportunity to illustrate some of the characteristics of the drug response curves in multiple dimensions that we cannot capture with handpicked features in the real data (for example the nonlinearity and change of direction in the feature space).

Here we give two analogous examples to Figure 3A using real data. In the first example we select 3 features and we plot 2 different classes with 3 compounds each in this hand-picked 3D space. The plot on the left shows the data before normalization, while the plot on the right shows the data after normalization.

To show the effect in the full feature space, we give a second example using PCA to plot in 3 dimensions (these figures have been included in the extended view). We plot 2 different classes with 3 compounds each in the first 3 PCA components. For the plot on the left we applied PCA to the data before normalization (data simply standardized), while for the plot on the right the we applied PCA to the normalized data.

The benefits of doing the normalisation were assessed by cross-validation accuracy on the training set. We show the effects of the normalization in individual features in our response to the last comment.

Line 185 - I am not familiar with the multi sensor fusion problem
They train and predict replicates? Then why did they create average fingerprints?

In our initial dataset, each compound contains a varying number of replicates for each tested dose. We create averaged fingerprints for each drug dose to smooth and balance the data. After the smoothing procedure, the dose replicates are replaced by bootstrapped averaged data points. Each compound contains a set number of averaged data points per dose. To perform cross-validation, we split the smoothed dataset, assigning some compounds of each class to the training fold and some to the test fold. We train the classifier using all the data points of the compounds of the training fold and then we make predictions for each data point in the test fold. To get a prediction at a compound-level, we use a majority vote procedure where each data point of the compound contributes one vote. The compound is predicted to belong to the class that gets the most votes.

In multi-sensor fusion, different classifiers are built for each condition and the predictions from each classifier provide a vote. In our case, that could translate to one classifier for each different dose if the doses were comparable. However, as explained in the text, different compounds have different potencies and a given dose for one compound is not equivalent from the same dose of another compound. Therefore, we cannot train separate classifiers. Instead, we use the voting procedure to the predictions of a single classifier for all the data points from different doses.

We have clarified the smoothing procedure adding an Extended View figure to illustrate its effect. Details can also be found in the methods.

They train on 60 compounds and test 16 compounds?

First, cross validation is performed on the 60 compounds to select features and model hyperparameters. The CV score achieved with the selected feature set and hyperparameters is reported in Figure 3B. Then, we use all 60 compounds in the training set to train a model with the selected features and hyperparameters. Finally, we predict the mode of action of the 16 compounds in the test set and report the result in Figure 3C.

Training data was used to determine an appropriate classifier - do they mean, "machine learning model"? How many machine learning models did they test - just the three mentioned models? They chose the multinomial logistic regression model based on cross-validation accuracy, with 1024 features?

We thoroughly tested three types of models: logistic regression, random forest and XGBoost. For each of these models we used the same methodology: using the training set only we performed feature selection and tuned the model hyperparameters as described in detail in the methods section. We finally selected the model/feature combination that gave us the highest cross validation accuracy. This was the multinomial logistic regression with 1024 features. Once we made this selection, we trained only this model/feature combination on the entire training set and made a prediction for the test set once.

There is no discussion on the multinomial regression model after that, why does it work on your data?

Despite being a linear classifier, logistic regression performs well in this classification problem probably because of the high dimensionality of the feature space, which renders the separation of classes with linear hyperplanes possible. At the same time, the use of linear boundaries limits overfitting, which is a big risk in high dimensional data with few samples (as discussed in a previous comment). In addition, it allows us to further control overfitting by using regularization and tuning the penalty parameter based on cross-validation.

We included this discussion in the methods section and added the essential point about the high dimensionality of the data in the results section.

Line 413: 3020 features + derived features and stimuli results in 9060 features per well.

Question: Which features are used where? Are the three stimuli used for the classification task?

We have clarified these points in the text and the methods, based on this and previous comments. As mentioned above, for the detection of compounds with significant effects (Figure 1D) we use all 9060 features. For the clustering in Figure 2, we concatenate the tierpsy_256 features from each condition (pre-stimulus, blue-light, post-stimulus). For the classification and novelty detection tasks, we perform feature selection starting from the full 9060-dimensional vector.

Line 443 we normalize each behavioral fingerprint to unit L2-norm. Rescaling in this way brings compounds with similar effect profiles across features but different potencies closer together in feature space.

Question: Can they give an example where compounds have similar effect profiles but different potencies?

In a previous comment, we have shown the effects of normalization to unit L2-norm on real data. Here, we will focus on the examples of compounds with similar effect but different potencies that are given in Figure 2C. To show the effect of the normalization, we have plotted some representative response curves of these compounds before and after normalization:

Spiroindolines:

Mitochondrial inhibitors:

The L2-norm brings the maximum effect observed in the response curves of different drugs closer together.

Reviewer #3:

Summary

The authors developed a pesticide drug classifier pipeline based on the effect drugs had on behavioral metrics in *C. elegans*. A variety of drugs were used with known targets to determine whether drugs with the same mode of action produced the same behavioral effects. Several drugs did indeed produce similar behavioral changes based on shared targets. The authors then leveraged this feature of their data to develop classifiers to predict a drug's mode of action based on its phenotypic effects. They furthered this analysis by comparing drugs with known targets, but differing sub-targets or off-targets, and showed certain drugs showed significantly different behavioral effects than other drugs that shared the same primary target, but different off-targets.

The key findings of this paper are that *C. elegans* can be used as a valuable symptomatic model for testing the mode-of-action of a variety of drugs. The authors have sufficient behavioral resolution to distinguish different categories of drugs, even though many of them cause motor defects in *C. elegans*. However, their detailed list of behavioral features enables them to distinguish the different classes of drugs.

Their work is greatly enabled by the high-throughput pipeline they have developed to test the

drugs. The liquid handling and small size of *C. elegans* allowed the researchers to test multiple drugs at different dosages and environmental conditions in a timely fashion.

General remarks

On the whole, I really enjoyed this paper. The authors have nicely extended the high-throughput behavioral analysis pipeline they have published in earlier work. As they mentioned, high-throughput symptomatic screening is challenging, especially if manual annotation is required. Their work has improved upon the standard pipeline by using machine vision, and a behavioral model that has limited complexity, but also a sufficient level of complexity to discriminate drug classes. People interested in drug development will be interested in this work, but also behavioral researchers using similar machine vision protocols to see potential applications that can be realized with these approaches.

Most of my comments are pretty minor:

1. What are the lines in Figure 1B? Confidence intervals, SD, quartiles?

The lines show the standard deviation. We have added this to the caption.

2. There appear to be far more drugs in Figure 2A than the 76 used for the classifiers. How were these drugs chosen? It appears there are 10 drug classes in Figure 2A, and 10 drug classes in Figure 3. Was there a bias for drugs that classified well in Figure 2A? If I missed how the 76 were chosen, I apologize. The method section describes how the training and test sets were split, but I don't understand how the original 76 compounds were chosen in the first place.

In Figure 2A, each row is a drug at a given dose so the total number of rows is 422 (76 compounds screened in at least 3 doses each). We have made this clearer in the figure caption and in the text in section '*Compounds with the same mode of action have similar effects on behavior*'.

From the starting library of 110 drugs, we dropped 17 with no detectable effect on behaviour. We have clarified this in the revised methods in section *Feature extraction and pre-processing*.

After dropping the 17 compounds we are left with 93 compounds with detectable effects. We then select the modes of action that are represented by at least 5 compounds and use only these classes for clustering and classification. These are the 10 classes in Figure 2A and Figure 3 and they include 76 compounds in total. These 10 classes were selected simply based on number of members (rather than classification accuracy or clustering potential) and therefore no bias was introduced in the classification results shown in Figure 3.

The classes that were not selected because they have fewer than 5 members (11 classes with 17 compounds in total) are assigned to the "novel" test set and are used to test the novelty detection algorithm.

3. I would have appreciated a more detailed explanation of the estimator they used, and what the final parameters were for their classifiers.

We initially tested three types of classifiers: logistic regression, random forest and XGBoost. Logistic regression performed better than the ensemble methods in terms of cross-validation

accuracy in the training set, so it was adopted for the entire pipeline. Despite being a linear classifier, logistic regression performs well in this classification problem probably because of the high dimensionality of the feature space, which renders the separation of classes with linear hyperplanes possible. At the same time, it limits overfitting both with the linear boundaries and the adoption of regularization. Finally, logistic regression has the advantage of readily providing class probabilities, which are used in subsequent analysis to derive novelty scores.

We included this discussing in the Methods section and added more details about the classifier in the Results section.

4. Is there raw data anywhere so readers can verify the analyses for themselves? This would be useful not only to verify the models, but also as a resource for the community.

Yes, we have uploaded the raw data and the tracking data to the Open Worm Movement Database on Zenodo. The links to the data are reported in Dataset EV3. We have also uploaded csv files with the feature summaries and corresponding metadata on Zenodo <https://doi.org/10.5281/zenodo.4681682>. The csv files can be used as source data to run the analysis scripts available on github (<https://github.com/Tierpsy/moaclassification>) and replicate all the results of the paper. We have added a Data Availability section in the paper where we list these sources in detail.

5. I think there should be a clarification that the dosage concentrations mentioned in the paper are not necessarily the dosage experienced by the animals themselves, but the concentration of drugs in the plates. The cuticle of the animals forms a notoriously difficult paper for many drugs. The availability of drugs to the animal may also be a source of variance in the data (which would contribute to differing dosage EC50s).

This is a good point and we have added it to the text where we talk about the dose effect including a reference to a recent review covering a range of xenobiotic defense mechanisms.

Hartman *et al.* (2021) Xenobiotic metabolism and transport in *Caenorhabditis elegans*, *Journal of Toxicology and Environmental Health, Part B*, DOI: 10.1080/10937404.2021.1884921

6. The following experiment isn't needed for the paper, but there are cuticle-mutant strains that have been used with better drug penetrance. For example, this paper:

<https://doi.org/10.1038/s41598-017-10454-3>

The usage of these strains may increase the number of drugs that have a behavioral effect on the animals, and/or decrease effective dosages.

We added a discussion of drug uptake and included this citation in the discussion. We are currently experimenting with more drug-susceptible strains and it will be interesting to see if increasing susceptibility also reduces variability and increases prediction accuracy.

I recommend this paper for publication.

Thank you for sending us your revised manuscript. We think that the performed revisions satisfactorily address the issues raised by the reviewers. As such, I am glad to inform you that we can soon accept the study for publication.

Before we can proceed with formally accepting the study, we would ask you to address some remaining editorial issues listed below.

2nd Authors' Response to Reviewers**21st Apr 2021**

The authors have made all requested editorial changes.

Thank you again for sending us your revised manuscript. We are now satisfied with the modifications made and I am pleased to inform you that your paper has been accepted for publication.

Corresponding Author Name: André Brown

Journal Submitted to: The EMBO Journal

Manuscript Number: MSB-2021-10267RR